# Small ReLU networks are powerful memorizers: a tight analysis of memorization capacity

**Chulhee Yun**
MIT
Cambridge, MA 02139
chulheey@mit.edu

**Suvrit Sra**
MIT
Cambridge, MA 02139
suvrit@mit.edu

**Ali Jadbabaie**
MIT
Cambridge, MA 02139
jadbabai@mit.edu

## Abstract

We study finite sample expressivity, i.e., memorization power of ReLU networks. Recent results require $N$ hidden nodes to memorize/interpolate arbitrary $N$ data points. In contrast, by exploiting depth, we show that 3-layer ReLU networks with $\Omega(\sqrt{N})$ hidden nodes can perfectly memorize most datasets with $N$ points. We also prove that width $\Theta(\sqrt{N})$ is *necessary and sufficient* for memorizing $N$ data points, proving tight bounds on memorization capacity. The sufficiency result can be extended to deeper networks; we show that an $L$-layer network with $W$ parameters in the hidden layers can memorize $N$ data points if $W = \Omega(N)$. Combined with a recent upper bound $O(WL \log W)$ on VC dimension, our construction is nearly tight for any fixed $L$. Subsequently, we analyze memorization capacity of residual networks under a general position assumption; we prove results that substantially reduce the known requirement of $N$ hidden nodes. Finally, we study the dynamics of stochastic gradient descent (SGD), and show that when initialized near a memorizing global minimum of the empirical risk, SGD quickly finds a nearby point with much smaller empirical risk.

## 1 Introduction

Recent results in deep learning indicate that over-parameterized neural networks can memorize arbitrary datasets [2, 53]. This phenomenon is closely related to the expressive power of neural networks, which have been long studied as universal approximators [12, 18, 21]. These results suggest that sufficiently large neural networks are expressive enough to fit any dataset perfectly.

With the widespread use of deep networks, recent works have focused on better understanding the power of depth [13, 17, 30, 33, 37, 38, 44, 45, 49, 50]. However, most existing results consider expressing *functions* (i.e., infinitely many points) rather than finite number of observations; thus, they do not provide a precise understanding the memorization ability of finitely large networks.

When studying finite sample memorization, several questions arise: Is a neural network capable of memorizing arbitrary datasets of a given size? How large must a neural network be to possess such capacity? These questions are the focus of this paper, and we answer them by studying *universal finite sample expressivity* and *memorization capacity*; these concepts are formally defined below.

**Definition 1.1.** We define (universal) **finite sample expressivity** of a neural network $f_{\boldsymbol{\theta}}(\cdot)$ (parametrized by $\boldsymbol{\theta}$) as the network's ability to satisfy the following condition:

> **For all** inputs $\{x_i\}_{i=1}^N \in \mathbb{R}^{d_x \times N}$ and **for all** $\{y_i\}_{i=1}^N \in [-1, +1]^{d_y \times N}$, **there exists** a parameter $\boldsymbol{\theta}$ such that $f_{\boldsymbol{\theta}}(x_i) = y_i$ for $1 \le i \le N$.

We define **memorization capacity** of a network to be the maximum value of $N$ for which the network has finite sample expressivity when $d_y = 1$.

Memorization capacity is related to, but is different from **VC dimension** of neural networks [3, 4]. Recall the definition of VC dimension of a neural network $f_{\boldsymbol{\theta}}(\cdot)$:

> The maximum value $N$ such that **there exists** a dataset $\{x_i\}_{i=1}^N \in \mathbb{R}^{d_x \times N}$ such that **for all** $\{y_i\}_{i=1}^N \in \{\pm 1\}^N$ **there exists** $\boldsymbol{\theta}$ such that $f_{\boldsymbol{\theta}}(x_i) = y_i$ for $1 \le i \le N$.

Notice that the key difference between memorization capacity and VC dimension is in the quantifiers in front of the $x_i$'s. Memorization capacity is always less than or equal to VC dimension, which means that an upper bound on VC dimension is also an upper bound on memorization capacity.

The study of finite sample expressivity and memorization capacity of neural networks has a long history, dating back to the days of perceptrons [6, 11, 22–24, 26, 36, 42, 48]; however, the older studies focus on shallow networks with traditional activations such as sigmoids, delivering limited insights for deep ReLU networks. Since the advent of deep learning, some recent results on modern architectures appeared, e.g., fully-connected neural networks (FNNs) [53], residual networks (ResNets) [20], and convolutional neural networks (CNNs) [35]. However, they impose assumptions on architectures that are neither practical nor realistic. For example, they require a hidden layer as wide as the number of data points $N$ [35, 53], or as many hidden nodes as $N$ [20], causing their theoretical results to be applicable only to very large neural networks; this can be unrealistic especially when $N$ is large.

## 1.1  Summary of our contributions

Before stating our contributions, a brief comment on "network size" is in order. The size of a neural network can be somewhat vague; it could mean width/depth, the number of edges, or the number of hidden nodes. We use "size" to refer to the number of hidden nodes in a network. This also applies to notions related to size; e.g., by a "small network" we mean a network with a small number of hidden nodes. For other measures of size such as width, we will use the words explicitly.

**1. Finite sample expressivity of neural networks.**   Our first set of results is on the finite sample expressivity of FNNs (Section 3), under the assumption of distinct data point $x_i$'s. For simplicity, we only summarize our results for ReLU networks, but they include hard-tanh networks as well.

- Theorem 3.1 shows that any 3-layer (i.e., 2-hidden-layer) ReLU FNN with hidden layer widths $d_1$ and $d_2$ can fit *any arbitrary* dataset if $d_1 d_2 \ge 4N d_y$, where $N$ is the number of data points and $d_y$ is the output dimension. For scalar outputs, this means $d_1 = d_2 = 2\sqrt{N}$ suffices to fit arbitrary data. This width requirement is significantly smaller than existing results on ReLU.

- The improvement is more dramatic for classification. If we have $d_y$ classes, Proposition 3.2 shows that a 4-layer ReLU FNN with hidden layer widths $d_1$, $d_2$, and $d_3$ can fit any dataset if $d_1 d_2 \ge 4N$ and $d_3 \ge 4d_y$. This means that $10^6$ data points in $10^3$ classes (e.g., ImageNet) can be memorized by a 4-layer FNN with hidden layer widths 2k-2k-4k.

- For $d_y = 1$, note that Theorem 3.1 shows a lower bound of $\Omega(d_1 d_2)$ on memorization capacity. We prove a matching upper bound in Theorem 3.3: we show that for shallow neural networks (2 or 3 layers), lower bounds on memorization capacity are tight.

- Proposition 3.4 extends Theorem 3.1 to deeper and/or narrower networks, and shows that if the sum of the number of edges between pairs of adjacent layers satisfies $d_{l_1} d_{l_1+1} + \cdots + d_{l_m} d_{l_m+1} = \Omega(N d_y)$, then universal finite sample expressivity holds. This gives a lower bound $\Omega(W)$ on memorization capacity, where $W$ is the number of edges in the network. Due to an upper bound $O(WL \log W)$ ($L$ is depth) on VC dimension [4], our lower bound is almost tight for fixed $L$.

Next, in Section 4, we focus on classification using ResNets; here $d_x$ denotes the input dimension and $d_y$ the number of classes. We assume here that data lies in general position.

- Theorem 4.1 proves that deep ResNets with $\frac{4N}{d_x} + 6d_y$ ReLU hidden nodes can memorize arbitrary datasets. Using the same proof technique, we also show in Corollary 4.2 that a 2-layer ReLU FNN can memorize arbitrary classification datasets if $d_1 \ge \frac{4N}{d_x} + 4d_y$. With the general position assumption, we can reduce the existing requirements of $N$ to a more realistic number.

**2.  Trajectory of SGD near memorizing global minima.**   Finally, in Section 5 we study the behavior of stochastic gradient descent (SGD) on the empirical risk of universally expressive FNNs.

- Theorem 5.1 shows that for *any* differentiable global minimum that memorizes, SGD initialized close enough (say $\epsilon$ away) to the minimum, quickly finds a point that has empirical risk $O(\epsilon^4)$

and is at most $2\epsilon$ far from the minimum. We emphasize that this theorem holds not only for memorizers explicitly constructed in Sections 3 and 4, but for *all* global minima that memorize. We note that we analyze without replacement SGD that is closer to practice than the simpler with-replacement version [19, 40]; thus, our analysis may be of independent interest in optimization.

## 1.2 Related work

**Universal finite sample expressivity of neural networks.** Literature on finite sample expressivity and memorization capacity of neural networks dates back to the 1960s. Earlier results [6, 11, 26, 36, 42] study memorization capacity of linear threshold networks.

Later, results on 2-layer FNNs with sigmoids [24] and other bounded activations [23] show that $N$ hidden nodes are sufficient to memorize $N$ data points. It was later shown that the requirement of $N$ hidden nodes can be improved by exploiting depth [22, 48]. Since these two works are highly relevant to our own results, we defer a detailed discussion/comparison until we present the precise theorems (see Sections 3.2 and 3.3).

With the advent of deep learning, there have been new results on modern activation functions and architectures. Zhang et al. [53] prove that one-hidden-layer ReLU FNNs with $N$ hidden nodes can memorize $N$ real-valued data points. Hardt and Ma [20] show that deep ResNets with $N + d_y$ hidden nodes can memorize arbitrary $d_y$-class classification datasets. Nguyen and Hein [35] show that deep CNNs with one of the hidden layers as wide as $N$ can memorize $N$ real-valued data points.

Soudry and Carmon [43] show that under a dropout noise setting, the training error is zero at every differentiable local minimum, for almost every dataset and dropout-like noise realization. However, this result is not comparable to ours because they assume that there is a multiplicative "dropout noise" at each hidden node and each data point. At $i$-th node of $l$-th layer, the slope of the activation function for the $j$-th data point is either $\epsilon_{i,l}^{(j)} \cdot 1$ (if input is positive) or $\epsilon_{i,l}^{(j)} \cdot s$ (if input is negative, $s \neq 0$), where $\epsilon_{i,l}^{(j)}$ is the multiplicative random (e.g., Gaussian) dropout noise. Their theorem statements hold for all realizations of these dropout noise factors *except a set of measure zero*. In contrast, our setting is free of these noise terms, and hence corresponds to a *specific* realization of such $\epsilon_{i,l}^{(n)}$'s.

**Convergence to global minima.** There exist numerous papers that study convergence of gradient descent or SGD to global optima of neural networks. Many previous results [9, 14, 29, 41, 46, 54, 55] study settings where data points are sampled from a distribution (e.g., Gaussian), and labels are generated from a "teacher network" that has the same architecture as the one being trained (i.e., realizability). Here, the goal of training is to recover the unknown (but fixed) true parameters. In comparison, we consider arbitrary datasets and networks, under a mild assumption (especially for overparametrized networks) that the network can memorize the data; the results are not directly comparable. Others [10, 47] study SGD on hinge loss under a bit strong assumption that the data is linearly separable.

Other recent results [1, 15, 16, 28, 58] focus on over-parameterized neural networks. In these papers, the widths of hidden layers are assumed to be huge, of polynomial order in $N$, such as $\Omega(N^4)$, $\Omega(N^6)$ or even greater. Although these works provide insights on how GD/SGD finds global minima easily, their width requirement is still far from being realistic.

A recent work [57] provides a mixture of observation and theory about convergence to global minima. The authors assume that networks can memorize the data, and that SGD follows a star-convex path to global minima, which they validate through experiments. Under these assumptions, they prove convergence of SGD to global minimizers. We believe our result is complementary: we provide sufficient conditions for networks to memorize the data, and our result does not assume anything about SGD's path but proves that SGD can find a point close to the global minimum.

**Remarks on generalization.** The ability of neural networks to memorize and generalize at the same time has been one of the biggest mysteries of deep learning [53]. Recent results on interpolation and "double descent" phenomenon indicate that memorization may not necessarily mean lack of generalization [5, 7, 8, 31, 32, 34]. We note that our paper focuses mainly on the ability of neural networks to memorize the training dataset, and that our results are separate from the discussion of generalization.

## 2 Problem setting and notation

In this section, we introduce the notation used throughout the paper. For integers $a$ and $b$, $a < b$, we denote $[a] := \{1, \ldots, a\}$ and $[a : b] := \{a, a+1, \ldots, b\}$. We denote $\{(x_i, y_i)\}_{i=1}^N$ the set of training data points, and our goal is to choose the network parameters $\boldsymbol{\theta}$ so that the network output $f_{\boldsymbol{\theta}}(x_i)$ is equal to $y_i$, for all $i \in [n]$. Let $d_x$ and $d_y$ denote input and output dimensions, respectively. Given input $x \in \mathbb{R}^{d_x}$, an $L$-layer fully-connected neural network computes output $f_{\boldsymbol{\theta}}(x)$ as follows:

$$
\begin{aligned}
a^0(x) &= x, \\
z^l(x) &= \boldsymbol{W}^l a^{l-1}(x) + \boldsymbol{b}^l, \quad a^l(x) = \sigma(z^l(x)), \quad \text{for } l \in [L-1], \\
f_{\boldsymbol{\theta}}(x) &= \boldsymbol{W}^L a^{L-1}(x) + \boldsymbol{b}^L.
\end{aligned}
$$

Let $d_l$ (for $l \in [L-1]$) denote the width of $l$-th hidden layer. For convenience, we write $d_0 := d_x$ and $d_L := d_y$. Here, $z^l \in \mathbb{R}^{d_l}$ and $a^l \in \mathbb{R}^{d_l}$ denote the input and output ($a$ for activation) of the $l$-th hidden layer, respectively. The output of a hidden layer is the entry-wise map of the input by the activation function $\sigma$. The bold-cased symbols denote parameters: $\boldsymbol{W}^l \in \mathbb{R}^{d_l \times d_{l-1}}$ is the weight matrix, and $\boldsymbol{b}^l \in \mathbb{R}^{d_l}$ is the bias vector. We define $\boldsymbol{\theta} := (\boldsymbol{W}^l, \boldsymbol{b}^l)_{l=1}^L$ to be the collection of all parameters. We write the network output as $f_{\boldsymbol{\theta}}(\cdot)$ to emphasize that it depends on parameters $\boldsymbol{\theta}$.

Our results in this paper consider piecewise linear activation functions. Among them, Sections 3 and 4 consider ReLU-like ($\sigma_R$) and hard-tanh ($\sigma_H$) activations, defined as follows:

$$
\sigma_R(t) := \begin{cases} s_+ t & t \geq 0, \\ s_- t & t < 0, \end{cases} \quad \sigma_H(t) := \begin{cases} -1 & t \leq -1, \\ t & t \in (-1, 1], \\ 1 & t > 1, \end{cases} = \frac{\sigma_R(t+1) - \sigma_R(t-1) - s_+ - s_-}{s_+ - s_-},
$$

where $s_+ > s_- \geq 0$. Note that $\sigma_R$ includes ReLU and Leaky ReLU. Hard-tanh activation ($\sigma_H$) is a piecewise linear approximation of tanh. Since $\sigma_H$ can be represented with two $\sigma_R$, any results on hard-tanh networks can be extended to ReLU-like networks with twice the width.

## 3 Finite sample expressivity of FNNs

In this section, we study universal finite sample expressivity of FNNs. For the training dataset, we make the following mild assumption that ensures consistent labels:

**Assumption 3.1.** *In the dataset $\{(x_i, y_i)\}_{i=1}^N$ assume that all $x_i$'s are distinct and all $y_i \in [-1, 1]^{d_y}$.*

### 3.1 Main results

We first state the main theorems on shallow FNNs showing tight lower and upper bounds on memorization capacity. Detailed discussion will follow in the next subsection.

**Theorem 3.1.** *Consider any dataset $\{(x_i, y_i)\}_{i=1}^N$ that satisfies Assumption 3.1. If*

- *a 3-layer hard-tanh FNN $f_{\boldsymbol{\theta}}$ satisfies $4 \lfloor d_1/2 \rfloor \lfloor d_2/(2d_y) \rfloor \geq N$; or*
- *a 3-layer ReLU-like FNN $f_{\boldsymbol{\theta}}$ satisfies $4 \lfloor d_1/4 \rfloor \lfloor d_2/(4d_y) \rfloor \geq N$,*

*then there exists a parameter $\boldsymbol{\theta}$ such that $y_i = f_{\boldsymbol{\theta}}(x_i)$ for all $i \in [N]$.*

Theorem 3.1 shows that if $d_1 d_2 = \Omega(N d_y)$ then we can memorize arbitrary datasets; this means that $\Omega(\sqrt{N d_y})$ hidden nodes are sufficient for memorization, in contrary to $\Omega(N d_y)$ requirements of recent results. By adding one more hidden layer, the next theorem shows that we can perfectly memorize any *classification* dataset using $\Omega(\sqrt{N} + d_y)$ hidden nodes.

**Proposition 3.2.** *Consider any dataset $\{(x_i, y_i)\}_{i=1}^N$ that satisfies Assumption 3.1. Assume that $y_i \in \{0, 1\}^{d_y}$ is the one-hot encoding of $d_y$ classes. Suppose one of the following holds:*

- *a 4-layer hard-tanh FNN $f_{\boldsymbol{\theta}}$ satisfies $4 \lfloor d_1/2 \rfloor \lfloor d_2/2 \rfloor \geq N$, and $d_3 \geq 2d_y$; or*
- *a 4-layer ReLU-like FNN $f_{\boldsymbol{\theta}}$ satisfies $4 \lfloor d_1/4 \rfloor \lfloor d_2/4 \rfloor \geq N$, and $d_3 \geq 4d_y$.*

*Then, there exists a parameter $\boldsymbol{\theta}$ such that $y_i = f_{\boldsymbol{\theta}}(x_i)$ for all $i \in [N]$.*

Notice that for scalar regression ($d_y = 1$), Theorem 3.1 proves a lower bound on memorization capacity of 3-layer neural networks: $\Omega(d_1 d_2)$. The next theorem shows that this bound is in fact *tight*.

**Theorem 3.3.** *Consider FNNs with $d_y = 1$ and piecewise linear activation $\sigma$ with $p$ pieces. If*

- *a 2-layer FNN $f_{\boldsymbol{\theta}}$ satisfies $(p-1)d_1 + 2 < N$; or*
- *a 3-layer FNN $f_{\boldsymbol{\theta}}$ satisfies $p(p-1)d_1d_2 + (p-1)d_2 + 2 < N$,*

*then there exists a dataset $\{(x_i, y_i)\}_{i=1}^N$ satisfying Assumption 3.1 such that for all $\boldsymbol{\theta}$, there exists $i \in [N]$ such that $y_i \neq f_{\boldsymbol{\theta}}(x_i)$.*

Theorems 3.1 and 3.3 together show **tight** lower and upper bounds $\Theta(d_1 d_2)$ on memorization capacity of 3-layer FNNs, which differ only in constant factors. Theorem 3.3 and the existing result on 2-layer FNNs [53, Theorem 1] also show that the memorization capacity of 2-layer FNNs is $\Theta(d_1)$.

**Proof ideas.** The proof of Theorem 3.1 is based on an intricate construction of parameters. Roughly speaking, we construct parameters that make each data point have its unique activation pattern in the hidden layers; more details are in Appendix B. The proof of Proposition 3.2 is largely based on Theorem 3.1. By assigning each class $j$ a unique real number $\rho_j$ (which is similar to the trick in Hardt and Ma [20]), we modify the dataset into a 1-D regression dataset; we then fit this dataset using the techniques in Theorem 3.1, and use the extra layer to recover the one-hot representation of the original $y_i$. Please see Appendix C for the full proof. The main proof idea of Theorem 3.3 is based on counting the number of "pieces" in the network output $f_{\boldsymbol{\theta}}(x)$ (as a function of $x$), inspired by Telgarsky [44]. For the proof, please see Appendix D.

## 3.2 Discussion

**Depth-width tradeoffs for finite samples.** Theorem 3.1 shows that if the two ReLU hidden layers satisfy $d_1 = d_2 = 2\sqrt{Nd_y}$, then the network can fit a given dataset perfectly. Proposition 3.2 is an improvement for classification, which shows that a 4-layer ReLU FNN can memorize any $d_y$-class classification data if $d_1 = d_2 = 2\sqrt{N}$ and $d_3 = 4d_y$.

As in other expressivity results, our results show that there are depth-width tradeoffs in the finite sample setting. For ReLU FNNs it is known that one hidden layer with $N$ nodes can memorize any scalar regression ($d_y = 1$) dataset with $N$ points [53]. By adding a hidden layer, the hidden node requirement is reduced to $4\sqrt{N}$, and Theorem 3.3 also shows that $\Theta(\sqrt{N})$ hidden nodes are *necessary and sufficient*. Ability to memorize $N$ data points with $N$ nodes is perhaps not surprising, because weights of each hidden node can be tuned to memorize a single data point. In contrast, the fact that width-$2\sqrt{N}$ networks can memorize is far from obvious; each hidden node must handle $\sqrt{N}/2$ data points on average, thus a more elaborate construction is required.

For $d_y$-class classification, by adding one more hidden layer, the requirement is improved from $4\sqrt{Nd_y}$ to $4\sqrt{N} + 4d_y$ nodes. This again highlights the power of depth in expressive power. Proposition 3.2 tells us that we can fit ImageNet[1] ($N \approx 10^6, d_y = 10^3$) with three ReLU hidden layers, using only 2k-2k-4k nodes. This "sufficient" size for memorization is surprisingly smaller (disregarding optimization aspects) than practical networks.

**Implications for ERM.** It is widely observed in experiments that deep neural networks can achieve zero empirical risk, but a concrete understanding of this phenomenon is still elusive. It is known that all local minima are global minima for empirical risk of linear neural networks [25, 27, 51, 52, 56], but this property fails to extend to nonlinear neural networks [39, 52]. This suggests that studying the gap between local minima and global minima could provide explanations for the success of deep neural networks. In order to study the gap, however, we have to know the risk value attained by global minima, which is already non-trivial even for shallow neural networks. In this regard, our theorems provide theoretical guarantees that even a shallow and narrow network can have zero empirical risk at global minima, *regardless of data and loss functions*—e.g., in a regression setting, for a 3-layer ReLU FNN with $d_1 = d_2 = 2\sqrt{Nd_y}$ there exists a global minimum that has zero empirical risk.

**The number of edges.** We note that our results do not contradict the common "insight" that at least $N$ edges are required to memorize $N$ data points. Our "small" network means a small number of hidden nodes, and it still has more than $N$ edges. The existing result [53] requires $(d_x + 2)N$ edges, while our construction for ReLU requires $4N + (2d_x + 6)\sqrt{N} + 1$ edges, which is much fewer.

**Relevant work on sigmoid.** Huang [22] proves that a 2-hidden-layer sigmoid FNNs with $d_1 = N/K + 2K$ and $d_2 = K$, where $K$ is a positive integer, can approximate $N$ arbitrary distinct data points. The author first partitions $N$ data points into $K$ groups of size $N/K$ each. Then, from the fact that the sigmoid function is strictly increasing and non-polynomial, it is shown that if the weights between input and first hidden layer is sampled *randomly*, then the output matrix of first hidden layer for each group is full rank with probability one. This is *not* the case for ReLU or hard-tanh, because they have "flat" regions in which rank could be lost. In addition, Huang [22] requires extra $2K$ hidden nodes in $d_1$ that serve as "filters" which let only certain groups of data points pass through. Our construction is not an extension of this result because we take a different strategy (Appendix B); we carefully choose parameters (instead of sampling) that achieve memorization with $d_1 = N/K$ and $d_2 = K$ (in hard-tanh case) without the need of extra $2K$ nodes, which enjoys a smaller width requirement and allows for more flexibility in the architecture. Moreover, we provide a converse result (Theorem 3.3) showing that our construction is rate-optimal in the number of hidden nodes.

## 3.3 Extension to deeper and/or narrower networks

What if the network is deeper than three layers and/or narrower than $\sqrt{N}$? Our next theorem shows that universal finite sample expressivity is not limited to 3-layer neural networks, and still achievable by exploiting depth even for narrower networks.

**Proposition 3.4.** *Consider any dataset $\{(x_i, y_i)\}_{i=1}^N$ that satisfies Assumption 3.1. For an L-layer FNN with hard-tanh activation ($\sigma_H$), assume that there exist indices $l_1, \ldots, l_m \in [L-2]$ that satisfy*

- $l_j + 1 < l_{j+1}$ *for* $j \in [m-1]$,
- $4 \sum_{j=1}^m \left\lfloor \frac{d_{l_j} - r_j}{2} \right\rfloor \left\lfloor \frac{d_{l_{j+1}} - r_j}{2 d_y} \right\rfloor \geq N$, *where* $r_j = d_y \mathbf{1}\{j > 1\} + \mathbf{1}\{j < m\}$, *for* $j \in [m]$,
- $d_k \geq d_y + 1$ *for all* $k \in \bigcup_{j \in [m-1]}[l_j + 2 : l_{j+1} - 1]$.
- $d_k \geq d_y$ *for all* $k \in [l_m + 2 : L - 1]$,

*where $\mathbf{1}\{\cdot\}$ is 0-1 indicator function. Then, there exists $\boldsymbol{\theta}$ such that $y_i = f_{\boldsymbol{\theta}}(x_i)$ for all $i \in [N]$.*

As a special case, note that for $L = 3$ (hence $m = 1$), the conditions boil down to that of Theorem 3.1. An immediate corollary of this fact is that the same result holds for ReLU(-like) networks with twice the width. Moreover, using the same proof technique as Proposition 3.2, this theorem can also be improved for classification datasets, by inserting one additional hidden layer between layer $l_m + 1$ and the output layer. Due to space limits, we defer the statement of these corollaries to Appendix A.

The proof of Proposition 3.4 is in Appendix E. We use Theorem 3.1 as a building block and construct a network (see Figure 2 in appendix) that fits a subset of dataset at each pair of hidden layers $l_j$–($l_j + 1$).

If any two adjacent hidden layers satisfy $d_l d_{l+1} = \Omega(N d_y)$, this network can fit $N$ data points ($m = 1$), even when all the other hidden layers have only one hidden node. Even with networks narrower than $\sqrt{N d_y}$ (thus $m > 1$), we can still achieve universal finite sample expressivity as long as there are $\Omega(N d_y)$ edges between disjoint pairs of adjacent layers. However, we have the "cost" $r_j$ in the width of hidden layers; this is because we fit subsets of the dataset using multiple pairs of layers. To do this, we need $r_j$ extra nodes to propagate input and output information to the subsequent layers. For more details, please refer to the proof.

Proposition 3.4 gives a lower bound $\Omega(\sum_{l=1}^{L-2} d_l d_{l+1})$ on memorization capacity for $L$-layer networks. For fixed input/output dimensions, this is indeed $\Omega(W)$, where $W$ is the number of edges in the network. On the other hand, Bartlett et al. [4] showed an upper bound $O(WL \log W)$ on VC dimension, which is also an upper bound on memorization capacity. Thus, for any fixed $L$, our lower bound is nearly tight. We conjecture that, as we have proved in 2- and 3-layer cases, the memorization capacity is $\Theta(W)$, independent of $L$; we leave closing this gap for future work.

For sigmoid FNNs, Yamasaki [48] claimed that a scalar regression dataset can be memorized if $d_x \lceil \frac{d_1}{2} \rceil + \lfloor \frac{d_1}{2} \rfloor \lceil \frac{d_2}{2} - 1 \rceil + \cdots + \lfloor \frac{d_{L-2}}{2} \rfloor \lceil \frac{d_{L-1}}{2} - 1 \rceil \geq N$. However, this claim was made under the stronger assumption of data lying in general position (see Assumption 4.1). Unfortunately, Yamasaki [48] does not provide a full proof of their claim, making it impossible to validate veracity of their construction (and we could not find their extended manuscript elsewhere).

# 4   Classification under the general position assumption

This section presents some results specialized in multi-class classification task under a slightly stronger assumption, namely the general position assumption. Since we are only considering classification in this section, we also assume that $y_i \in \{0, 1\}^{d_y}$ is the one-hot encoding of $d_y$ classes.

**Assumption 4.1.** *For a finite dataset $\{(x_i, y_i)\}_{i=1}^{N}$, assume that no $d_x + 1$ data points lie on the same affine hyperplane. In other words, the data point $x_i$'s are in general position.*

We consider **residual networks** (ResNets), defined by the following architecture:

$$h^0(x) = x,$$
$$h^l(x) = h^{l-1}(x) + \boldsymbol{V}^l \sigma(\boldsymbol{U}^l h^{l-1}(x) + \boldsymbol{b}^l) + \boldsymbol{c}^l, \ l \in [L-1],$$
$$g_{\boldsymbol{\theta}}(x) = \boldsymbol{V}^L \sigma(\boldsymbol{U}^L h^{L-1}(x) + \boldsymbol{b}^L) + \boldsymbol{c}^L,$$

which is similar to the previous work by Hardt and Ma [20], except for extra bias parameters $\boldsymbol{c}^l$. In this model, we denote the number hidden nodes in the $l$-th residual layer as $d_l$; e.g., $\boldsymbol{U}^l \in \mathbb{R}^{d_l \times d_x}$.

We now present a theorem showing that any dataset can be memorized with small ResNets.

**Theorem 4.1.** *Consider any dataset $\{(x_i, y_i)\}_{i=1}^{N}$ that satisfies Assumption 4.1. Assume also that $d_x \geq d_y$. Suppose one of the following holds:*

- *a hard-tanh ResNet $g_{\boldsymbol{\theta}}$ satisfies $\sum_{l=1}^{L-1} d_l \geq \frac{2N}{d_x} + 2d_y$ and $d_L \geq d_y$; or*

- *a ReLU-like ResNet $g_{\boldsymbol{\theta}}$ satisfies $\sum_{l=1}^{L-1} d_l \geq \frac{4N}{d_x} + 4d_y$ and $d_L \geq 2d_y$.*

*Then, there exists $\boldsymbol{\theta}$ such that $y_i = g_{\boldsymbol{\theta}}(x_i)$ for all $i \in [N]$.*

The previous work by Hardt and Ma [20] proves universal finite sample expressivity using $N + d_y$ hidden nodes (i.e., $\sum_{l=1}^{L-1} d_l \geq N$ and $d_L \geq d_y$) for ReLU activation, under the assumption that $x_i$'s are distinct *unit* vectors. Note that neither this assumption nor Assumption 4.1 implies the other; however, our assumption is quite mild in the sense that for any given dataset, adding small random Gaussian noise to $x_i$'s makes the dataset satisfy the assumption, with probability 1.

The main idea for the proof is that under the general position assumption, for any choice of $d_x$ points there exists an affine hyperplane that contains *only* these $d_x$ points. Each hidden node can choose $d_x$ data points and "push" them to the right direction, making perfect classification possible. We defer the details to Appendix F.1. Using the same technique, we can also prove an improved result for 2-layer (1-hidden-layer) FNNs. The proof of the following corollary can be found in Appendix F.2.

**Corollary 4.2.** *Consider any dataset $\{(x_i, y_i)\}_{i=1}^{N}$ that satisfies Assumption 4.1. Suppose one of the following holds:*

- *a 2-layer hard-tanh FNN $f_{\boldsymbol{\theta}}$ satisfies $d_1 \geq \frac{2N}{d_x} + 2d_y$; or*

- *a 2-layer ReLU-like FNN $f_{\boldsymbol{\theta}}$ satisfies $d_1 \geq \frac{4N}{d_x} + 4d_y$.*

*Then, there exists $\boldsymbol{\theta}$ such that $y_i = f_{\boldsymbol{\theta}}(x_i)$ for all $i \in [N]$.*

Our results show that under the general position assumption, perfect memorization is possible with only $\Omega(N/d_x + d_y)$ hidden nodes rather than $N$, in both ResNets and 2-layer FNNs. Considering that $d_x$ is typically in the order of hundreds or thousands, our results reduce the hidden node requirements down to more realistic network sizes. For example, consider CIFAR-10 dataset: $N = 50,000$, $d_x = 3,072$, and $d_y = 10$. Previous results require at least 50k ReLUs to memorize this dataset, while our results require 126 ReLUs for ResNets and 106 ReLUs for 2-layer FNNs.

# 5   Trajectory of SGD near memorizing global minima

In this section, we study the behavior of without-replacement SGD near memorizing global minima.

We restrict $d_y = 1$ for simplicity. We use the same notation as defined in Section 2, and introduce here some additional definitions. We assume that each activation function $\sigma$ is piecewise linear with at least two pieces (e.g., ReLU or hard-tanh). Throughout this section, we slightly abuse the notation $\boldsymbol{\theta}$ to denote the concatenation of vectorizations of all the parameters $(\boldsymbol{W}^l, \boldsymbol{b}^l)_{l=1}^{L}$.

We are interested in minimizing the empirical risk $\mathfrak{R}(\boldsymbol{\theta})$, defined as the following:

$$\mathfrak{R}(\boldsymbol{\theta}) := \frac{1}{N} \sum\nolimits_{i=1}^{N} \ell(f_{\boldsymbol{\theta}}(x_i); y_i),$$

where $\ell(z; y) : \mathbb{R} \mapsto \mathbb{R}$ is the loss function parametrized by $y$. We assume the following:

**Assumption 5.1.** *The loss function $\ell(z; y)$ is a strictly convex and three times differentiable function of $z$. Also, for any $y$, there exists $z \in \mathbb{R}$ such that $z$ is a global minimum of $\ell(z; y)$.*

Assumption 5.1 on $\ell$ is satisfied by standard losses such as squared error loss. Note that logistic loss does not satisfy Assumption 5.1 because the global minimum is not attained by any finite $z$.

Given the assumption on $\ell$, we now formally define the **memorizing** global minimum.

**Definition 5.1.** A point $\boldsymbol{\theta}^*$ is a memorizing global minimum of $\mathfrak{R}(\cdot)$ if $\ell'(f_{\boldsymbol{\theta}^*}(x_i); y_i) = 0, \forall i \in [N]$.

By convexity, $\ell'(f_{\boldsymbol{\theta}^*}(x_i); y_i) = 0$ for all $i$ implies that $\mathfrak{R}(\boldsymbol{\theta})$ is (globally) minimized at $\boldsymbol{\theta}^*$. Also, existence of a memorizing global minimum of $\mathfrak{R}$ implies that all global minima are memorizing.

Although $\ell$ is a differentiable function of $z$, the empirical risk $\mathfrak{R}(\boldsymbol{\theta})$ is not necessarily differentiable in $\boldsymbol{\theta}$ because we are using piecewise linear activations. In this paper, we only consider differentiable points of $\mathfrak{R}(\cdot)$; since nondifferentiable points lie in a set of measure zero and SGD never reaches such points in reality, this is a reasonable assumption.

We consider minimizing the empirical risk $\mathfrak{R}(\boldsymbol{\theta})$ using without-replacement mini-batch SGD. We use $B$ as mini-batch size, so it takes $E := N/B$ steps to go over $N$ data points in the dataset. For simplicity we assume that $N$ is a multiple of $B$. At iteration $t = kE$, it partitions the dataset at random, into $E$ sets of cardinality $B$: $B^{(kE)}, B^{(kE+1)}, \ldots, B^{(kE+E-1)}$, and uses these sets to estimate gradients. After each epoch (one pass through the dataset), the data is "reshuffled" and a new partition is used. Without-replacement SGD is known to be more difficult to analyze than with-replacement SGD (see [19, 40] and references therein), although more widely used in practice.

More concretely, our SGD algorithm uses the update rule $\boldsymbol{\theta}^{(t+1)} \leftarrow \boldsymbol{\theta}^{(t)} - \eta g^{(t)}$, where we fix the step size $\eta$ to be a constant throughout the entire run and $g^{(t)}$ is the gradient estimate

$$g^{(t)} = \frac{1}{B} \sum\nolimits_{i \in B^{(t)}} \ell'(f_{\boldsymbol{\theta}^{(t)}}(x_i); y_i) \nabla_{\boldsymbol{\theta}} f_{\boldsymbol{\theta}^{(t)}}(x_i).$$

For each $k$, $\bigcup_{t=kE}^{kE+E-1} B^{(t)} = [N]$. Note also that if $B = N$, we recover vanilla gradient descent.

Now consider a memorizing global minimum $\boldsymbol{\theta}^*$. We define vectors $\nu_i := \nabla_{\boldsymbol{\theta}} f_{\boldsymbol{\theta}^*}(x_i)$ for all $i \in [N]$. We can then express any iterate $\boldsymbol{\theta}^{(t)}$ of SGD as $\boldsymbol{\theta}^{(t)} = \boldsymbol{\theta}^* + \boldsymbol{\xi}^{(t)}$, and then further decompose the "perturbation" $\boldsymbol{\xi}^{(t)}$ as the sum of two orthogonal components $\boldsymbol{\xi}_{\|}^{(t)}$ and $\boldsymbol{\xi}_{\perp}^{(t)}$, where $\boldsymbol{\xi}_{\|}^{(t)} \in \text{span}(\{\nu_i\}_{i=1}^{N})$ and $\boldsymbol{\xi}_{\perp}^{(t)} \in \text{span}(\{\nu_i\}_{i=1}^{N})^{\perp}$. Also, for a vector $v$, let $\|v\|$ denote its $\ell_2$ norm.

## 5.1 Main results and discussion

We now state the main theorem of the section. For the proof, please refer to Appendix G.

**Theorem 5.1.** *Suppose a memorizing global minimum $\boldsymbol{\theta}^*$ of $\mathfrak{R}(\boldsymbol{\theta})$ is given, and that $\mathfrak{R}(\cdot)$ is differentiable at $\boldsymbol{\theta}^*$. Then, there exist positive constants $\rho$, $\gamma$, $\lambda$, and $\tau$ satisfying the following: if initialization $\boldsymbol{\theta}^{(0)}$ satisfies $\|\boldsymbol{\xi}^{(0)}\| \leq \rho$, then*

$$\mathfrak{R}(\boldsymbol{\theta}^{(0)}) - \mathfrak{R}(\boldsymbol{\theta}^*) = O(\|\boldsymbol{\xi}^{(0)}\|^2),$$

*and SGD with step size $\eta < \gamma$ satisfies*

$$\|\boldsymbol{\xi}_{\|}^{(kE+E)}\| \leq (1 - \eta\lambda)\|\boldsymbol{\xi}_{\|}^{(kE)}\|, \quad \text{and} \quad \|\boldsymbol{\xi}^{(kE+E)}\| \leq \|\boldsymbol{\xi}^{(kE)}\| + \eta\lambda\|\boldsymbol{\xi}_{\|}^{(kE)}\|,$$

*as long as $\|\boldsymbol{\xi}_{\|}^{(t)}\| \geq \tau\|\boldsymbol{\xi}^{(t)}\|^2$ holds for all $t \in [kE, kE + E - 1]$. As a consequence, at the **first** iterate $t^* \geq 0$ where the condition $\|\boldsymbol{\xi}_{\|}^{(t)}\| \geq \tau\|\boldsymbol{\xi}^{(t)}\|^2$ is **violated**, we have*

$$\|\boldsymbol{\xi}^{(t^*)}\| \leq 2\|\boldsymbol{\xi}^{(0)}\|, \quad \text{and} \quad \mathfrak{R}(\boldsymbol{\theta}^{(t^*)}) - \mathfrak{R}(\boldsymbol{\theta}^*) \leq C\|\boldsymbol{\xi}^{(0)}\|^4,$$

*for some positive constant $C$.*

The full description of constants $\rho$, $\gamma$, $\lambda$, $\tau$, and $C$ can be found in Appendix G. They are dependent on a number of terms, such as $N$, $B$, the Taylor expansions of loss $\ell(f_{\boldsymbol{\theta}^*}(x_i); y_i)$ and network output $f_{\boldsymbol{\theta}^*}(x_i)$ around the memorizing global minimum $\boldsymbol{\theta}^*$, maximum and minimum strictly positive eigenvalues of $H = \sum_{i=1}^{N} \ell''(f_{\boldsymbol{\theta}^*}(x_i); y_i)\nu_i \nu_i^T$. The constant $\rho$ must be small enough so that as long as $\|\boldsymbol{\xi}\| \leq \rho$, the slopes of piecewise linear activation functions evaluated for data points $x_i$ do not change from $\boldsymbol{\theta}^*$ to $\boldsymbol{\theta}^* + \boldsymbol{\xi}$.

Notice that for small perturbation $\boldsymbol{\xi}$, the Taylor expansion of network output $f_{\boldsymbol{\theta}^*}(x_i)$ is written as $f_{\boldsymbol{\theta}^*+\boldsymbol{\xi}}(x_i) = f_{\boldsymbol{\theta}^*}(x_i) + \nu_i^T \boldsymbol{\xi}_{\|} + O(\|\boldsymbol{\xi}\|^2)$, because $\nu_i \perp \boldsymbol{\xi}_{\perp}$ by definition. From this perspective, Theorem 5.1 shows that if initialized near global minima, the component in the perturbation $\boldsymbol{\xi}$ that induces first-order perturbation of $f_{\boldsymbol{\theta}^*}(x_i)$, namely $\boldsymbol{\xi}_{\|}$, decays exponentially fast until SGD finds a nearby point that has much smaller risk ($O(\|\boldsymbol{\xi}^{(0)}\|^4)$) than the initialization ($O(\|\boldsymbol{\xi}^{(0)}\|^2)$). Note also that our result is completely deterministic, and independent of the partitions of the dataset taken by the algorithm; the theorem holds true even if the algorithm is not "stochastic" and just cycles through the dataset in a fixed order without reshuffling.

We would like to emphasize that Theorem 5.1 holds for *any* memorizing global minima of FNNs, not only for the ones explicitly constructed in Sections 3 and 4. Moreover, the result is not dependent on the network size or data distribution. As long as the global minimum memorizes the data, our theorem holds *without* any depth/width requirements or distributional assumptions, which is a noteworthy difference that makes our result hold in more realistic settings than existing ones.

The remaining question is: what happens after $t^*$? Unfortunately, if $\|\boldsymbol{\xi}_{\|}^{(t)}\| \leq \tau \|\boldsymbol{\xi}^{(t)}\|^2$, we cannot ensure exponential decay of $\|\boldsymbol{\xi}_{\|}^{(t)}\|$, especially if it is small. Without exponential decay, one cannot show an upper bound on $\|\boldsymbol{\xi}^{(t)}\|$ either. This means that after $t^*$, SGD may even diverge or oscillate near global minimum. Fully understanding the behavior of SGD after $t^*$ seems to be a more difficult problem, which we leave for future work.

## 6 Conclusion and future work

In this paper, we show that fully-connected neural networks (FNNs) with $\Omega(\sqrt{N})$ nodes are expressive enough to perfectly memorize $N$ arbitrary data points, which is a significant improvement over the recent results in the literature. We also prove the converse stating that at least $\Theta(\sqrt{N})$ nodes are necessary; these two results together provide tight bounds on memorization capacity of neural networks. We further extend our expressivity results to deeper and/or narrower networks, providing a nearly tight bound on memorization capacity for these networks as well. Under an assumption that data points are in general position, we prove that classification datasets can be memorized with $\Omega(N/d_x + d_y)$ hidden nodes in deep residual networks and one-hidden-layer FNNs, reducing the existing requirement of $\Omega(N)$. Finally, we study the dynamics of stochastic gradient descent (SGD) on empirical risk, and showed that if SGD is initialized near a global minimum that perfectly memorizes the data, it quickly finds a nearby point with small empirical risk. Several future topics are open; e.g., 1) tight bounds on memorization capacity for deep FNNs and other architectures, 2) deeper understanding of SGD dynamics in the presence of memorizing global minima.

#### Acknowledgments

We thank Alexander Rakhlin for helpful discussion. All the authors acknowledge support from DARPA Lagrange. Chulhee Yun also thanks Korea Foundation for Advanced Studies for their support. Suvrit Sra also acknowledges support from an NSF-CAREER grant and an Amazon Research Award.

## Footnotes

[1]after omitting the inconsistently labeled items

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
