[Supplementary Material]

# A Deferred theorem statements

In this section, we state the theorems that were omitted in Section 3.3 due to lack of space. First, we start by stating the ReLU-like version of Theorem 3.4:

**Corollary A.1.** *Consider any dataset $\{(x_i, y_i)\}_{i=1}^N$ that satisfies Assumption 3.1. For an L-layer FNN with ReLU(-like) activation ($\sigma_R$), assume that there exist indices $l_1, \ldots, l_m \in [L-2]$ that satisfies*

- $l_j + 1 < l_{j+1}$ *for* $j \in [m-1]$,
- $4 \sum_{j=1}^m \left\lfloor \frac{d_{l_j} - r_j}{4} \right\rfloor \left\lfloor \frac{d_{l_{j+1}} - r_j}{4d_y} \right\rfloor \geq N$, *where* $r_j = d_y \mathbf{1}\{j > 1\} + \mathbf{1}\{j < m\}$, *for* $j \in [m]$,
- $d_k \geq d_y + 1$ *for all* $k \in \bigcup_{j \in [m-1]} [l_j + 2 : l_{j+1} - 1]$.
- $d_k \geq d_y$ *for all* $k \in [l_m + 2 : L - 1]$,

*where* $\mathbf{1}\{\cdot\}$ *is 0-1 indicator function. Then, there exists $\boldsymbol{\theta}$ such that $y_i = f_{\boldsymbol{\theta}}(x_i)$ for all $i \in [N]$.*

The idea is that anything that holds for hard-tanh activation holds for ReLU networks that has double the width. One difference to note is that the number of nodes needed for "propagating" input and output information (the circle and diamond nodes in Figure 2) has not doubled. This is because merely propagating the information without nonlinear distortion can be done with a single ReLU-like activation.

The next corollaries are special cases for classification. One can check that with $L = 4$ and $m = 2$ (hence $l_1 = 1$ and $l_2 = 3$), these boil down to Proposition 3.2.

**Corollary A.2.** *Consider any dataset $\{(x_i, y_i)\}_{i=1}^N$ that satisfies Assumption 3.1. Assume that $y_i \in \{0, 1\}^{d_y}$ is the one-hot encoding of $d_y$ classes. For an L-layer FNN with hard-tanh activation ($\sigma_H$), assume that there exist indices $l_1, \ldots, l_m \in [L-1]$ ($m \geq 2$) that satisfies*

- $l_j + 1 < l_{j+1}$ *for* $j \in [m-1]$,
- $4 \sum_{j=1}^{m-1} \left\lfloor \frac{d_{l_j} - r_j}{2} \right\rfloor \left\lfloor \frac{d_{l_{j+1}} - r_j}{2} \right\rfloor \geq N$, *where* $r_j = \mathbf{1}\{j > 1\} + \mathbf{1}\{j < m-1\}$, *for* $j \in [m-1]$,
- $d_{l_m} \geq 2d_y$,
- $d_k \geq 2$ *for all* $k \in \bigcup_{j \in [m-2]} [l_j + 2 : l_{j+1} - 1]$.
- $d_k \geq d_y$ *for all* $k \in [l_m + 1 : L - 1]$.

*Then, there exists $\boldsymbol{\theta}$ such that $y_i = f_{\boldsymbol{\theta}}(x_i)$ for all $i \in [N]$.*

**Corollary A.3.** *Consider any dataset $\{(x_i, y_i)\}_{i=1}^N$ that satisfies Assumption 3.1. Assume that $y_i \in \{0, 1\}^{d_y}$ is the one-hot encoding of $d_y$ classes. For an L-layer FNN with ReLU(-like) activation ($\sigma_R$), assume that there exist indices $l_1, \ldots, l_m \in [L-1]$ ($m \geq 2$) that satisfies*

- $l_j + 1 < l_{j+1}$ *for* $j \in [m-1]$,
- $4 \sum_{j=1}^{m-1} \left\lfloor \frac{d_{l_j} - r_j}{4} \right\rfloor \left\lfloor \frac{d_{l_{j+1}} - r_j}{4} \right\rfloor \geq N$, *where* $r_j = \mathbf{1}\{j > 1\} + \mathbf{1}\{j < m-1\}$, *for* $j \in [m-1]$,
- $d_{l_m} \geq 4d_y$,
- $d_k \geq 2$ *for all* $k \in \bigcup_{j \in [m-2]} [l_j + 2 : l_{j+1} - 1]$.
- $d_k \geq d_y$ *for all* $k \in [l_m + 1 : L - 1]$.

*Then, there exists $\boldsymbol{\theta}$ such that $y_i = f_{\boldsymbol{\theta}}(x_i)$ for all $i \in [N]$.*

The proof of Corollaries A.2 and A.3 can be done by easily combining the ideas in proofs of Proposition 3.2 and Proposition 3.4, hence omitted.

# B Proof of Theorem 3.1

We prove the theorem by constructing a parameter $\boldsymbol{\theta}$ that perfectly fits the dataset. We will prove the theorem for hard-tanh ($\sigma_H$) only, because extension to ReLU-like ($\sigma_R$) is straightforward from its

**Figure 1.** Illustration of the construction for $d_1 = d_2 = 4$. Each box corresponds to a hidden node with hard-tanh activation. In each hidden node, the numbers written in the three parts are indices of data points that are clipped to $-1$ at output (left), those clipped to $+1$ (right), and those unchanged (center). One can check for all indices that outputs of layer 2 sum to $y_i + 1$.

definition. To convey the main idea more clearly, we first prove the theorem for $d_y = 1$, and later discuss how to extend to $d_y > 1$.

For a data point $x_i$, the corresponding input and output of the $l$-th hidden layer is written as $z^l(x_i)$ and $a^l(x_i)$, respectively. Moreover, $z^l_j(x_i)$ and $a^l_j(x_i)$ denote the input and output of the $j$-th node of the $l$-th hidden layer. For weight matrices $\boldsymbol{W}^l$, we will denote its $(j, k)$-th entry as $\boldsymbol{W}^l_{j,k}$, its $j$-th row as $\boldsymbol{W}^l_{j,:}$, and its $j$-th column as $\boldsymbol{W}^l_{:,j}$. Similarly, $\boldsymbol{b}^l_j$ denote the $j$-th component of the bias vector $\boldsymbol{b}^l$. To simplify notation, we will denote $p := d_1$ and $q := d_2$, for the rest of the proof. Assume for simplicity that $p$ is a multiple of 2, $q$ is a multiple of 2, and $pq = N$.

### B.1 Proof sketch

The proof consists of three steps, one for each layer. In this subsection, we will describe each step in the following three paragraphs. Then, the next three subsections will provide the full details of each step.

In the first step, we down-project all input data points to a line, using a random vector $u \in \mathbb{R}^{d_x}$. Different $x_i$'s are mapped to different $u^T x_i$'s, so we have $N$ distinct $u^T x_i$'s on the line. Now re-index the data points in increasing order of $u^T x_i$, and divide total $N$ data points into $p$ groups with $q$ points each. To do this, each row $\boldsymbol{W}^1_{j,:}$ of $\boldsymbol{W}^1$ is chosen as $u^T$ multiplied by a scalar. We choose the appropriate scalar for $\boldsymbol{W}^1_{j,:}$ and bias $\boldsymbol{b}^1_j$, so that the input to the $j$-th hidden node in layer 1, $z^1_j(\cdot)$, satisfies the following: (1) $z^1_j(x_i) \in (-1, 1)$ for indices $i \in [jq - q + 1 : jq]$, and (2) $z^1_j(x_i) \in (-1, 1)^c$ for all other indices so that they are "clipped" by $\sigma_{\mathrm{H}}$.

In the second step, for each hidden node in layer 2, we pick one point each from these $p$ groups and map their values to desired $y_i$. More specifically, for $k$-th node in layer 2, we define an index set $\mathcal{I}_k$

(with cardinality $p$) that contains exactly one element from each $[jq - q + 1 : jq]$, and choose $\boldsymbol{W}_{k,:}^2$ and $\boldsymbol{b}_k^2$ such that $z_k^2(x_i) = y_i$ for $i \in \mathcal{I}_k$ and $z_k^2(x_i) \in [-1, 1]^c$ for $i \notin \mathcal{I}_k$. This is possible because for each $k$, we are solving $p$ linear equations with $p + 1$ variables.

As we will see in the details, the first and second steps involve alternating signs and a carefully designed choice of index sets $\mathcal{I}_k$ so that sum of output $a_k^2(\cdot)$ of each node in layer 2 becomes $y_i + 1$. Figure 1 shows a simple illustration for $p = q = 4$. With this choice, we can make the output $f_{\boldsymbol{\theta}}(x_i)$ become simply $y_i$ for all $i \in [N]$, thereby perfectly memorizing the dataset.

## B.2 Input to layer 1: down-project and divide

First, recall from Assumption 3.1 that all $x_i$'s are distinct. This means that for any pair of data points $x_i$ and $x_{i'}$, the set of vectors $u \in \mathbb{R}^{d_x}$ satisfying $u^T x_i = u^T x_{i'}$ has measure zero. Thus, if we sample any $u$ from some distribution (e.g., Gaussian), $u$ satisfies $u^T x_i \neq u^T x_{i'}$ for all $i \neq i'$ with probability 1. This is a standard proof technique also used in other papers; please see e.g., Huang [22, Lemma 2.1].

We choose any such $u$, and without loss of generality, re-index the data points in increasing order of $u^T x_i$: $u^T x_1 < u^T x_2 < \cdots < u^T x_N$. Now define $c_i := u^T x_i$ for all $i \in [N]$, and additionally, $c_0 = c_1 - \delta$ and $c_{N+1} = c_N + \delta$, for any $\delta > 0$.

Now, we are going to define $\boldsymbol{W}^1$ and $\boldsymbol{b}^1$ such that the input to the $j$-th ($j \in [p]$) hidden node in layer 1 has $z_j^1(x_i) \in (-1, 1)$ for indices $i \in [jq - q + 1 : jq]$, and $z_j^1(x_i) \in (-1, 1)^c$ for any other points. We also alternate the order of data points, which will prove useful in later steps. More concretely, we define the $j$-th row of $\boldsymbol{W}^1$ and $j$-th component of $\boldsymbol{b}^1$ to be

$$\boldsymbol{W}_{j,:}^1 = (-1)^{j-1} \frac{4}{c_{jq} + c_{jq+1} - c_{jq-q} - c_{jq-q+1}} u^T,$$

$$\boldsymbol{b}_j^1 = (-1)^j \frac{c_{jq} + c_{jq+1} + c_{jq-q} + c_{jq-q+1}}{c_{jq} + c_{jq+1} - c_{jq-q} - c_{jq-q+1}}.$$

When $j$ is odd, it is easy to check that $z_j^1(\cdot)$ satisfies

$$-1 < z_j^1(x_{jq-q+1}) < \cdots < z_j^1(x_{jq}) < +1,$$
$$z_j^1(x_i) < -1 \text{ for } i \leq jq - q,$$
$$z_j^1(x_i) > +1 \text{ for } i > jq,$$

so that the output $a_j^1(\cdot)$ satisfies

$$-1 < a_j^1(x_{jq-q+1}) < \cdots < a_j^1(x_{jq}) < +1, \tag{1}$$
$$a_j^1(x_i) = -1 \text{ for } i \leq jq - q, \tag{2}$$
$$a_j^1(x_i) = +1 \text{ for } i > jq. \tag{3}$$

When $j$ is even, by a similar argument:

$$+1 > a_j^1(x_{jq-q+1}) > \cdots > a_j^1(x_{jq}) > -1, \tag{4}$$
$$a_j^1(x_i) = +1 \text{ for } i \leq jq - q, \tag{5}$$
$$a_j^1(x_i) = -1 \text{ for } i > jq. \tag{6}$$

## B.3 Layer 1 to 2: place at desired positions

At each node of layer 2, we will show how to place $p$ points at the right position, and the rest of points in the clipping region. After that, we will see that adding up all node outputs of layer 2 gives $y_i + 1$ for all $i$.

For $k$-th hidden node in layer 2 ($k \in [q]$), define a set

$$\mathcal{I}_k := \{k, 2q + 1 - k, 2q + k, 4q + 1 - k, \ldots, pq + 1 - k\}.$$

Note that $|\mathcal{I}_k| = p$. Also, let us denote the elements of $\mathcal{I}_k$ as $i_{k,1}, \ldots, i_{k,p}$ in increasing order. For example, $i_{k,1} = k$, $i_{k,2} = 2q + 1 - k$, and so on. We can see that $i_{k,j} \in [jq - q + 1 : jq]$.

For each $k$, our goal is to construct $\boldsymbol{W}_{k,:}^2$ and $\boldsymbol{b}_k^2$ so that the input to the $k$-th node of layer 2 places data points indexed with $i \in \mathcal{I}_k$ to the desired position $y_i \in [-1, 1]$, and the rest of data points $i \notin \mathcal{I}_k$ outside $[-1, 1]$.

**Case 1: odd $k$.**    We first describe how to construct $\boldsymbol{W}^2_{k,:}$ and $\boldsymbol{b}^2_k$ for **odd** $k$'s. First of all, consider data points $x_{i_{k,j}}$'s in $\mathcal{I}_k$. We want to choose parameters so that the input to the $k$-th node is equal to $y_{i_{k,j}}$'s:

$$z^2_k(x_{i_{k,j}}) = \sum\nolimits_{l=1}^{p} \boldsymbol{W}^2_{k,l} a^1_l(x_{i_{k,j}}) + \boldsymbol{b}^2_k = y_{i_{k,j}},$$

for all $j \in [p]$. This is a system of $p$ linear equations with $p+1$ variables, which can be represented in a matrix-vector product form:

$$M_k \begin{bmatrix} (\boldsymbol{W}^2_{k,:})^T \\ \boldsymbol{b}^2_k \end{bmatrix} = \begin{bmatrix} y_{i_{k,1}} \\ \vdots \\ y_{i_{k,p}} \end{bmatrix}, \tag{7}$$

where the $(j,l)$-th entry of matrix $M_k \in \mathbb{R}^{p \times (p+1)}$ is defined by $a^1_l(x_{i_{k,j}})$ for $j \in [p]$ and $l \in [p]$, and $(j, p+1)$-th entries are all equal to 1.

With the matrix $M_k$ defined from the above equation, we state the lemma whose simple proof is deferred to Appendix H for better readability:

**Lemma B.1.** *For any $k \in [q]$, the matrix $M_k \in \mathbb{R}^{p \times (p+1)}$ satisfies the following properties:*
1. *$M_k$ has full column rank.*
2. *There exists a vector $\nu \in \mathrm{null}(M_k)$ such that the first $p$ components of $\nu$ are all strictly positive.*

Lemma B.1 implies that for any $y_{i_{k,1}}, \ldots, y_{i_{k,p}}$, there exist infinitely many solutions $(\boldsymbol{W}^2_{k,:}, \boldsymbol{b}^2_k)$ for (7) of the form $\mu + \alpha\nu$, where $\mu$ is any particular solution satisfying the linear system and $\alpha$ is any scalar. This means that by scaling $\alpha$, and we can make $\boldsymbol{W}^2_{k,:}$ as large as we want, without hurting $z^2_k(x_i) = y_i$ for $i \in \mathcal{I}_k$.

It is now left to make sure that any other data points $i \notin \mathcal{I}_k$ have $z^2_k(x_i) \in [-1,1]^c$. As we will show, this can be done by making $\alpha > 0$ sufficiently large.

Now fix any odd $j \in [p]$, and consider $i_{k,j} \in \mathcal{I}_k$, and recall $i_{k,j} \in [jq - q + 1 : jq]$. Fix any other $i \in [jq - q + 1 : i_{k,j} - 1]$. By Eqs (2), (3), (5) and (6), the output of $l$-th node in layer 1 ($l \neq j$) is the same for $i$ and $i_{k,j}$: $a^1_l(x_i) = a^1_l(x_{i_{k,j}})$.

In contrast, for $a^1_j(\cdot)$, we have $a^1_j(x_i) < a^1_j(x_{i_{k,j}})$ (1). Since $z^2_k(x_{i_{k,j}}) = \sum_l \boldsymbol{W}^2_{k,l} a^1_l(x_{i_{k,j}}) + \boldsymbol{b}^2_k = y_{i_{k,j}}$, large enough $\boldsymbol{W}^2_{k,j} > 0$ will make $z^2_k(x_i) < -1$, resulting in $a^2_k(x_i) = -1$; the output for $x_i$ is clipped. A similar argument can be repeated for $i \in [i_{k,j} + 1 : jq]$, so that for large enough $\boldsymbol{W}^2_{k,j} > 0$,

$$a^2_k(x_i) = -1, \ \forall i \in [jq - q + 1 : i_{k,j} - 1]$$
$$a^2_k(x_i) = +1, \ \forall i \in [i_{k,j} + 1 : jq].$$

Similarly, for even $j \in [p]$, large $\boldsymbol{W}^2_{k,j} > 0$ will make

$$a^2_k(x_i) = +1, \ \forall i \in [jq - q + 1 : i_{k,j} - 1]$$
$$a^2_k(x_i) = -1, \ \forall i \in [i_{k,j} + 1 : jq].$$

Summarizing, for large enough $\boldsymbol{W}^2_{k,:} > 0$ (achieved by making $\alpha > 0$ large), the output of the $k$-th node of layer 2 satisfies $a^2_k(x_i) = y_i$, $\forall i \in \mathcal{I}_k$, and

$$a^2_k(x_i) = -1, \ \forall i \in \bigcup\nolimits_{\substack{j \in [0:p] \\ j \text{ even}}} [i_{k,j} + 1 : i_{k,j+1} - 1], \tag{8}$$

$$a^2_k(x_i) = +1, \ \forall i \in \bigcup\nolimits_{\substack{j \in [p] \\ j \text{ odd}}} [i_{k,j} + 1 : i_{k,j+1} - 1], \tag{9}$$

where $i_{k,0} := 0$ and $i_{k,p+1} := N + 1$ for all $k \in [q]$.

**Case 2: even $k$.** For even $k$'s, we can repeat the same process, except that we push $\alpha < 0$ to large negative number, so that $W^2_{k,:} < 0$ is sufficiently large negative. By following a very similar argument, we can make the output of the $k$-th node of layer 2 satisfy $a^2_k(x_i) = y_i, \forall i \in \mathcal{I}_k$, and

$$a^2_k(x_i) = +1, \forall i \in \bigcup_{\substack{j \in [0:p] \\ j \text{ even}}} [i_{k,j} + 1 : i_{k,j+1} - 1], \tag{10}$$

$$a^2_k(x_i) = -1, \forall i \in \bigcup_{\substack{j \in [p] \\ j \text{ odd}}} [i_{k,j} + 1 : i_{k,j+1} - 1]. \tag{11}$$

## B.4 Layer 2 to output: add them all

Quite surprisingly, adding up $a^2_k(x_i)$ for all $k \in [q]$ gives $y_i + 1$ for all $i \in [N]$. To prove this, first observe that the index sets $\mathcal{I}_1, \mathcal{I}_2, \ldots, \mathcal{I}_q$ form a partition of $[N]$. So, proving $\sum_{l=1}^q a^2_l(x_{i_{k,j}}) = y_{i_{k,j}} + 1$ for all $j \in [p]$ and $k \in [q]$ suffices.

By the definition of $i_{k,1} = k, i_{k,2} = 2q + 1 - k, i_{k,3} = 2q + k, \ldots, i_{k,p-1} = (p-2)q + k, i_{k,p} = pq + 1 - k$, we can see the following chains of inequalities:

$$jq - q + 1 = i_{1,j} < i_{2,j} < \cdots < i_{q,j} = jq \text{ for } j \text{ odd},$$
$$jq - q + 1 = i_{q,j} < \cdots < i_{2,j} < i_{1,j} = jq \text{ for } j \text{ even}.$$

Fix any $k \in [q]$, and any odd $j \in [p]$. From the above chains of inequalities, we can observe that

$$i_{k,j} \in [i_{l,j} + 1 : i_{l,j+1} - 1] \text{ if } l < k,$$
$$i_{k,j} \in [i_{l,j-1} + 1 : i_{l,j} - 1] \text{ if } l > k.$$

Now, for $x_{i_{k,j}}$, we will sum up $a^2_l(x_{i_{k,j}})$ for $l \in [q]$. First, for $1 \le l < k$, we have $i_{k,j} \in [i_{l,j} + 1 : i_{l,j+1} - 1]$. Since $j$ is odd, from Eqs (9) and (11),

$$a^2_l(x_{i_{k,j}}) = \begin{cases} +1 & \text{for odd } l < k, \\ -1 & \text{for even } l < k. \end{cases}$$

Similarly, for $k < l \le w$, we have $i_{k,j} \in [i_{l,j-1} + 1 : i_{l,j} - 1]$. Since $j$ is odd, from Eqs (8) and (10),

$$a^2_l(x_{i_{k,j}}) = \begin{cases} -1 & \text{for odd } l > k, \\ +1 & \text{for even } l > k. \end{cases}$$

Then, the sum over $l \ne k$ always results in $+1$, so

$$\sum_{l=1}^q a^2_l(x_{i_{k,j}}) = y_{i_{k,j}} + \sum_{l \ne k} a^2_l(x_{i_{k,j}}) = y_{i_{k,j}} + 1.$$

For any fixed even $j \in [p]$, we can similarly prove the same thing. We have

$$i_{k,j} \in [i_{l,j-1} + 1 : i_{l,j} - 1] \text{ if } l < k,$$
$$i_{k,j} \in [i_{l,j} + 1 : i_{l,j+1} - 1] \text{ if } l > k,$$

for even $j$. From this point, the remaining steps are exactly identical to the odd case.

Now that we know $\sum_{l=1}^q a^2_l(x_i) = y_i + 1$, we can choose $W^3 = \mathbf{1}_q^T$ and $b^3 = -1$ so that $f_{\boldsymbol{\theta}}(x_i) = y_i$. This finishes the proof of Theorem 3.1 for $d_y = 1$.

## B.5 Proof for $d_y > 1$

The proof for $d_y > 1$ is almost the same. Assume that $p := d_1$ is a multiple of 2, $q := d_2$ is a multiple of $2d_y$, and $pq = Nd_y$. Now partition the nodes in the 2nd layer into $d_y$ groups of size $q/d_y$. For each of the $d_y$ groups, we can do the exact same construction as done in $d_y = 1$ case, to fit each coordinate of $y_i$ perfectly. This is possible because we can share $a^1(x_i)$ for fitting different components of $y_i$.

## C  Proof of Proposition 3.2

For the proof, we will abuse the notation slightly and let $y_i \in [d_y]$ denote the class that $x_i$ belongs to. The idea is simple: assign distinct real numbers $\rho_1, \ldots, \rho_{d_y}$ to each of the $d_y$ classes, define a new 1-dimensional regression dataset $\{(x_i, \rho_{y_i})\}_{i=1}^N$, and do the construction in Theorem 3.1 up to layer 2 for the new dataset. Then, we have $\sum_{l=1}^{d_2} a_l^2(x_i) = \rho_{y_i} + 1$, as seen in the proof of Theorem 3.1.

Now, at layer 3, consider the following "gate" activation function $\sigma_G$, which allows values in $(-1, +1)$ to "pass," while blocking others. This can be implemented with two $\sigma_H$'s or four $\sigma_R$'s:

$$\sigma_G(t) := \begin{cases} t+1 & -1 \le t \le 0, \\ -t+1 & 0 \le t \le 1, \\ 0 & \text{otherwise.} \end{cases} = \tfrac{1}{2}(\sigma_H(2t+1) + \sigma_H(-2t+1)).$$

For each class $j \in [d_y]$, we can choose appropriate parameters to implement a gate that allows $\rho_j$ to "pass" the gate, while blocking any other $\rho_{j'}$, $j' \ne j$. The output of the gate is then connected to the $j$-th output node of the network. This way, we can perfectly recover the one-hot representation for each data point.

## D  Proof of Theorem 3.3

Our proof is based on the idea of counting the number of pieces of piecewise linear functions by Telgarsky [44]. Consider any vector $u \in \mathbb{R}^{d_x}$, and define the following dataset: $x_i = iu$, $y_i = (-1)^i$, for all $i \in [N]$.

With piecewise linear activation functions, the network output $f_\theta(x)$ is also a piecewise affine function of $x$. If we define $\bar{f}_\theta(t) := f_\theta(tu)$, $\bar{f}_\theta(t)$ must have at least $N-1$ linear pieces to be able to fit the given dataset $\{(x_i, y_i)\}_{i=1}^N$. We will prove the theorem by counting the maximum number of linear pieces in $\bar{f}_\theta(t)$.

We will use the following lemma, which is a slightly improved version of Telgarsky [44, Lemma 2.3]:

**Lemma D.1.** *If $g : \mathbb{R} \mapsto \mathbb{R}$ and $h : \mathbb{R} \mapsto \mathbb{R}$ are piecewise linear with $k$ and $l$ linear pieces, respectively, then $g + h$ is piecewise linear with at most $k + l - 1$ pieces, and $g \circ h$ is piecewise linear with at most $kl$ pieces.*

For proof of the lemma, please refer to Telgarsky [44].

Consider the output of layer 1 $\bar{a}^1(t) := a^1(tu)$, restricted for $x = tu$. For each $j \in [d_1]$, $\bar{a}_j^1(\cdot)$ has at most $p$ pieces. The input to layer 2 is a weighted sum of $\bar{a}_j^1(\cdot)$'s, so each $\bar{z}_k^2(t) := z_k^2(tu)$ has $(p-1)d_1 + 1$ pieces, resulting in maximum $p(p-1)d_1 + p$ pieces in the corresponding output $\bar{a}_k^2(t)$. Again, the weighted sum of $d_2$ such $\bar{a}_k^2(\cdot)$'s have at most $(p(p-1)d_1 + p - 1)d_2 + 1 = p(p-1)d_1d_2 + (p-1)d_2 + 1$ pieces.

From this calculation, we can see that the output of a 2-layer network has at most $(p-1)d_1 + 1$ pieces, and a 3-layer network has $p(p-1)d_1d_2 + (p-1)d_2 + 1$. If these number of pieces are strictly smaller than $N-1$, the network can never perfectly fit the given dataset.

## E  Proof of Proposition 3.4

For Proposition 3.4, we will use the network from Theorem 3.1 as a building block to construct the desired parameters. The parameters we construct will result in a network illustrated in Figure 2. Please note that the arrows are drawn for *nonzero* parameters only, and all the missing arrows just mean that the parameters are zero. We are not using a special architecture; we are still in the full connected network regime.

In the proof of Theorem 3.1, we down-projected $x_i$'s to $u^T x_i =: c_i$, and fitted $c_1, \ldots, c_N$ to corresponding $y_1, \ldots, y_N$. Then, what happens outside the range of the dataset? Recall from Section B.2 that we defined $c_0 := c_1 - \delta$ and $c_{N+1} := c_N + \delta$ for $\delta > 0$ and constructed $\boldsymbol{W}^1$ and $\boldsymbol{b}^1$ using them. If we go back to the proof of Theorem 3.1, we can check that if $u^T x \le c_0$ or $u^T x \ge c_{N+1}$, $a_k^2(x) = -1$ for odd $k$'s and $+1$ for even $k$'s, resulting in $\sum_{k=1}^q a_k^2(x) = 0$ for all such $x$'s. For a quick check, consider imaginary indices 0 and 17 in Figure 1 and see which sides (left or right) of the 2nd-layer hidden nodes they will be written.

**Figure 2.** Illustration of network parameter construction in Proposition 3.4. The circle/diamond nodes represent those carrying input/output information, respectively. The rectangular blocks are groups of nodes across two layers whose parameters are constructed from Theorem 3.1 to fit data points.

Now consider partitioning $N$ data points into $m$ subsets of cardinalities $N_1, \ldots, N_m$ in the following way. We first down-project the data to get $u^T x_i$'s, and re-index data points in increasing order of $u^T x_i$'s. The first $N_1$ points go into the first subset, the next $N_2$ to the second, and so on. Then, consider constructing $m$ separate networks (by Theorem 3.1) such that each network fits each subset, *except that we let $\boldsymbol{b}^3 = \boldsymbol{0}$*. As seen above, the sum of the outputs of *all* these $m$ networks will be $y_i + \boldsymbol{1}$, for all $i \in [N]$. Thus, by fitting subsets of dataset separately and summing together, we can still memorize $N$ data points.

The rest of the proof can be explained using Figure 2. For simplicity, we assume that

- For all $j \in [m]$, $d_{l_j} - r_j$ is a multiple of 2, and $d_{l_j+1} - r_j$ is a multiple of $2d_y$,
- $\sum_{j=1}^{m}(d_{l_j} - r_j)(d_{l_j+1} - r_j) = Nd_y$,
- $d_k = 1$ for all $k \in [l_1 - 1]$,
- $d_k = d_y + 1$ for all $k \in \bigcup_{j \in [m-1]}[l_j + 2 : l_{j+1} - 1]$,
- $d_k = d_y$ for all $k \in [l_m + 2 : L - 1]$.

Also, let $N_j := (d_{l_j} - r_j)(d_{l_j+1} - r_j)/d_y$ for $j \in [m]$.

From the input layer to layer 1, we down-project $x_i$'s using a random vector $u$, and scale $\boldsymbol{W}^1 := u^T$ and choose $\boldsymbol{b}^1$ appropriately so that $\boldsymbol{W}^1 x_i + \boldsymbol{b}^1 \in (-1, +1)$ for all $i \in [N]$. As seen in the circle nodes in Figure 2, this "input information" will be propagated up to layer $l_m - 1$ to provide input data needed for fitting.

At layer $l_j - 1$, the weights and bias into the rectangular block across layers $l_j$–$(l_j + 1)$ is selected in the same way as Section B.2. Inside each block, the subset of $N_j$ data points are fitted using the construction of Theorem 3.1, but this time we fit to $\frac{y_i - 1}{2}$ instead of $y_i$, in order to make sure that output information is not clipped by hard-tanh. The output of $(l_j + 1)$-th layer nodes in the block are added up and connected to diamond nodes in layer $l_j + 2$. For the $N_j$ data points in the subset, the input to the diamond nodes will be $\frac{y_i + \boldsymbol{1}}{2}$ (instead of $y_i + \boldsymbol{1}$), and $\boldsymbol{0}$ for any other data points. As seen in Figure 2, this output information is propagated up to the output layer.

After fitting all $m$ subsets, the output value of diamond nodes at layer $L - 1$ is $\frac{y_i + \boldsymbol{1}}{2}$, for all $i$. We can scale and shift this value at the output layer and get $y_i = f_{\boldsymbol{\theta}}(x_i)$.

## F   Proofs of Theorem 4.1 and Corollary 4.2

### F.1   Proof of Theorem 4.1

The key observation used in the proof is that due to the general position assumption, if we pick any $d_x$ data points in the same class, then there always exists an affine hyperplane that contains exactly

these $d_x$ points. This way, we can pick $d_x$ data points per hidden node and "push" them far enough to specific directions (depending on the classes), so that the last hidden layer can distinguish the classes based on the location of data points.

We use $N_k$ to denote the number of data points in class $k \in [d_y]$. Also, for $k \in [d_y]$, let $x_{(k)}^{\max}$ be the maximum value of the $k$-th component of $x_i$ over all $i \in [N]$. Also, let $e_k$ be the $k$-th standard unit vector in $\mathbb{R}^{d_x}$.

Now, consider the gate activation function $\sigma_{\mathrm{G}}$, which was also used in the proof of Proposition 3.2 (Appendix C). This activation allows values in $(-1, +1)$ to "pass," while blocking others. This can be implemented with two hard-tanh ($\sigma_{\mathrm{H}}$) functions or four ReLU-like ($\sigma_{\mathrm{R}}$) functions:

$$\sigma_{\mathrm{G}}(t) := \begin{cases} t+1 & -1 \leq t \leq 0, \\ -t+1 & 0 \leq t \leq 1, \\ 0 & \text{otherwise.} \end{cases} = \tfrac{1}{2}(\sigma_{\mathrm{H}}(2t+1) + \sigma_{\mathrm{H}}(-2t+1)).$$

Up to layer $L-1$, for now we will assume that the activation at the hidden nodes is $\sigma_{\mathrm{G}}$. We will later count the actual number of hard-tanh or ReLU-like nodes required.

For class $k \in [d_y]$, we use $\lceil \frac{N_k}{d_x} \rceil$ gate hidden nodes for class $k$. Each hidden node picks and pushes $d_x$ data points in class $k$ far enough to the direction of $e_k$. Each data point is chosen only once. Suppose that the hidden node is the $j$-th hidden node in $l$-th layer ($l \in [L-1], j \in [d_l]$). Pick $d_x$ data points in class $k$ that are not yet "chosen," then there is an affine hyperplane $u^T x + c = 0$ that contains only these points.

Using the activation $\sigma_{\mathrm{G}}$, we can make the hidden node have output 1 for the chosen $d_x$ data points and 0 for all remaining data points. This can be done by setting the incoming parameters

$$\boldsymbol{U}_{j,:}^l = \alpha u^T, \quad \boldsymbol{b}_j^l = \alpha c,$$

where $\alpha > 0$ is a big enough positive constant so that $|\alpha(u^T x_i + c)| > 1$ and thus $\sigma_{\mathrm{G}}(\alpha(u^T x_i + c)) = 0$ for all unpicked data points $x_i$. Then, choose the outgoing parameters

$$\boldsymbol{V}_{:,j}^l = \beta e_k, \quad \boldsymbol{c}^l = \mathbf{0}$$

where $\beta > 0$ will be specified shortly. Notice that since each data point is chosen only once, the $d_x$ data points were never chosen previously. Therefore, for these $d_x$ data points, we have

$$h^j(x_i) = x_i, \qquad \text{for } j \in [l-1], \text{ and}$$
$$h^j(x_i) = x_i + \beta e_k, \text{ for } j \in [l : L-1],$$

because they will never be chosen again by other hidden nodes. We choose big enough $\beta$ to make sure that the $k$-th component of $h^l(x_i)$ (i.e., $h_k^l(x_i)$) is bigger than $x_{(k)}^{\max} + 1$. We also determine $\beta$ carefully so that adding $\beta e_k$ does not break the general position assumption. The values of $\beta$ that breaks the general position lie in a set of measure zero, so we can sample $\beta$ from some suitable continuous random distribution to avoid this.

After doing this to all data points, $h^{L-1}(x_i)$ satisfies the following property: For $x_i$'s that are in class $k$, $h_k^{L-1}(x_i) \geq x_{(k)}^{\max} + 1$, and for $x_i$'s that are not in class $k$, $h_k^{L-1}(x_i) \leq x_{(k)}^{\max}$.

At layer $L$, by assumption we have $d_L \geq d_y$ in case of hard-tanh ResNet. We assume $d_L = d_y$ for simplicity, and choose

$$\boldsymbol{U}^L = \begin{bmatrix} 2 \cdot I_{d_y \times d_y} & \mathbf{0}_{d_y \times (d_x - d_y)} \end{bmatrix}, \quad \boldsymbol{b}^L = \begin{bmatrix} -2x_{(1)}^{\max} - 1 \\ -2x_{(2)}^{\max} - 1 \\ \vdots \\ -2x_{(d_y)}^{\max} - 1 \end{bmatrix},$$

then by clipping of hard-tanh, for $x_i$ in class $k$, the $k$-th component of $\sigma(\boldsymbol{U}^L h^{L-1}(x_i) + \boldsymbol{b}^L)$ is $+1$ and all the other components are $-1$. Now, by choosing

$$\boldsymbol{V}^L = \frac{1}{2} \cdot I_{d_y \times d_y}, \quad \boldsymbol{c}^L = \frac{1}{2} \mathbf{1}_{d_y},$$

we can recover the one-hot representation: $g_{\boldsymbol{\theta}}(x_i) = y_i$, for all $i \in [N]$. For ReLU-like ResNets, we can do the same job by using $d_L = 2d_y$.

Finally, let us count the number of hidden nodes used, for layers up to $L-1$. Recall that we use $\lceil \frac{N_k}{d_x} \rceil$ **gate** activation nodes for class $k$. Note that the total number of gate activations used is bounded above by

$$\sum_{k=1}^{d_y} \left\lceil \frac{N_k}{d_x} \right\rceil \leq \sum_{k=1}^{d_y} \left( \frac{N_k}{d_x} + 1 \right) = \frac{N}{d_x} + d_y,$$

and each gate activation can be constructed with two hard-tanh nodes or four ReLU-like nodes. Therefore, $\sum_{l=1}^{L-1} d_l \geq \frac{2N}{d_x} + 2d_y$ and $d_L \geq d_y$ is the sufficient condition for a hard-tanh ResNet to realize the above construction, and ReLU-like ResNets require twice as many hidden nodes.

### F.2 Proof of Corollary 4.2

The main idea of the proof is exactly the same. We use $\lceil \frac{N_k}{d_x} \rceil$ gate activation nodes for class $k$, and choose $d_x$ data points in the same class per each hidden node. When the hidden node is the $j$-th node in the hidden layer and the chosen points are from class $k$, we choose

$$\boldsymbol{W}^2_{:,j} = \boldsymbol{e}_k, \; \boldsymbol{b}^2 = \boldsymbol{0}.$$

This way, one can easily recover the one-hot representation and achieve $f_{\boldsymbol{\theta}}(x_i) = y_i$.

## G   Proof of Theorem 5.1

The outline of the proof is as follows. Recall that we write $\boldsymbol{\theta}^{(t)}$ as $\boldsymbol{\theta}^* + \boldsymbol{\xi}^{(t)}$. By the chain rule, we have

$$\nabla_{\boldsymbol{\theta}} \mathfrak{R}(\boldsymbol{\theta}^* + \boldsymbol{\xi}^{(t)}) = \frac{1}{N} \sum_{i=1}^{N} \ell'(f_{\boldsymbol{\theta}^*+\boldsymbol{\xi}^{(t)}}(x_i); y_i) \nabla_{\boldsymbol{\theta}} f_{\boldsymbol{\theta}^*+\boldsymbol{\xi}^{(t)}}(x_i).$$

If $\boldsymbol{\xi}^{(t)}$ is small enough, the terms $\ell'(f_{\boldsymbol{\theta}^*+\boldsymbol{\xi}^{(t)}}(x_i); y_i)$ and $\nabla_{\boldsymbol{\theta}} f_{\boldsymbol{\theta}^*+\boldsymbol{\xi}^{(t)}}(x_i)$ can be expressed in terms of perturbation on $\ell'(f_{\boldsymbol{\theta}^*}(x_i); y_i)$ and $\nabla_{\boldsymbol{\theta}} f_{\boldsymbol{\theta}^*}(x_i)$, respectively (Lemma G.1). We then use the lemma and prove each statement of the theorem.

We first begin by introducing more definitions and symbols required for the proof. As mentioned in the main text, we'll abuse the notation $\boldsymbol{\theta}$ to mean the concatenation of vectorizations of all the parameters $(\boldsymbol{W}^l, \boldsymbol{b}^l)_{l=1}^{L}$. To simplify the notation, we define $\ell_i(\boldsymbol{\theta}) := \ell(f_{\boldsymbol{\theta}}(x_i); y_i)$. Same thing applies for derivatives of $\ell$: $\ell'_i(\boldsymbol{\theta}) := \ell'(f_{\boldsymbol{\theta}}(x_i); y_i)$, and so on.

Now, for each data point $i \in [N]$ and each layer $l \in [L-1]$, define the following diagonal matrix:

$$J^l_{\boldsymbol{\theta}}(x_i) := \text{diag}\left( \begin{bmatrix} \sigma'(z^l_1(x_i)) & \cdots & \sigma'(z^l_{d_l}(x_i)) \end{bmatrix} \right) \in \mathbb{R}^{d_l \times d_l},$$

where $\sigma'$ is the derivative of the activation function $\sigma$, wherever it exists.

Now consider a memorizing global minimum $\boldsymbol{\theta}^*$. As done in the main text, we will express any other point $\boldsymbol{\theta}$ as $\boldsymbol{\theta} = \boldsymbol{\theta}^* + \boldsymbol{\xi}$, where $\boldsymbol{\xi}$ is the vectorized version of perturbations. By assumption, $\mathfrak{R}(\cdot)$ is differentiable at $\boldsymbol{\theta}^*$; this means that $J^l_{\boldsymbol{\theta}^*}(x_i)$ are well-defined at $\boldsymbol{\theta}^*$ for all data points and layers $l \in [L-1]$. Moreover, since $\sigma$ is piecewise linear, there exists a small enough positive constant $\rho_c$ such that for any $\boldsymbol{\xi}$ satisfying $\|\boldsymbol{\xi}\| \leq \rho_c$, the slopes of activation functions stay constant, i.e., $J^l_{\boldsymbol{\theta}^*+\boldsymbol{\xi}}(x_i) = J^l_{\boldsymbol{\theta}^*}(x_i)$ for all $i \in [N]$ and $l \in [L-1]$.

Now, as in the main text, define vectors $\nu_i := \nabla_{\boldsymbol{\theta}} f_{\boldsymbol{\theta}^*}(x_i)$ for all $i \in [N]$. We can then express $\boldsymbol{\xi}$ as the sum of two orthogonal components $\boldsymbol{\xi}_{\|}$ and $\boldsymbol{\xi}_{\perp}$, where $\boldsymbol{\xi}_{\|} \in \text{span}(\{\nu_i\}_{i=1}^{N})$ and $\boldsymbol{\xi}_{\perp} \in \text{span}(\{\nu_i\}_{i=1}^{N})^{\perp}$. We also define $P_{\nu}$ to be the projection matrix onto $\text{span}(\{\nu_i\}_{i=1}^{N})$; note that $\boldsymbol{\xi}_{\|} = P_{\nu} \boldsymbol{\xi}$.

Using the fact that perturbations are small, we can calculate the deviation of network output $f_{\boldsymbol{\theta}^*+\boldsymbol{\xi}}(x_i)$ from $f_{\boldsymbol{\theta}^*}(x_i)$, and use Taylor expansion of $\ell$ and $\ell'$ to show the following lemma, whose proof is deferred to Appendix I.

**Lemma G.1.** *For any given memorizing global minimum $\boldsymbol{\theta}^*$ of $\mathfrak{R}(\cdot)$, there exist positive constants $\rho_s$ $(\leq \rho_c)$, $C_1$, $C_2$, $C_3$, $C_4$, and $C_5$ such that, if $\|\boldsymbol{\xi}\| \leq \rho_s$, the following holds for all $i \in [N]$:*

$$\ell_i(\boldsymbol{\theta}^* + \boldsymbol{\xi}) - \ell_i(\boldsymbol{\theta}^*) \leq C_1(C_2\|\boldsymbol{\xi}_{\|}\| + C_3\|\boldsymbol{\xi}\|^2)^2,$$

$$\ell_i'(\boldsymbol{\theta}^* + \boldsymbol{\xi}) = \ell_i''(\boldsymbol{\theta}^*)\nu_i^T\boldsymbol{\xi}_{\|} + R_i(\boldsymbol{\xi}),$$

$$\nabla_{\boldsymbol{\theta}} f_{\boldsymbol{\theta}^* + \boldsymbol{\xi}}(x_i) = \nu_i + \mu_i(\boldsymbol{\xi}),$$

*where the remainder/perturbation terms satisfy*

$$|R_i(\boldsymbol{\xi})| \leq C_4\|\boldsymbol{\xi}\|^2, \text{ and } \|\mu_i(\boldsymbol{\xi})\| \leq C_5\|\boldsymbol{\xi}\|.$$

Besides the constants defined in Lemma G.1, define

$$C_6 := \max_{i \in [N]} \ell_i''(\boldsymbol{\theta}^*)\|\nu_i\|.$$

Also, it will be shown in the proof of Lemma G.1 that $C_2 := \max_{i \in [N]} \|\nu_i\|$. Given Lemma G.1, we are now ready to prove Theorem 5.1.

Let us first consider the case where all $\nu_i$'s are zero vectors, so $\text{span}(\{\nu_i\}_{i=1}^N) = \{\mathbf{0}\}$. For such a pathological case, $\boldsymbol{\xi}_{\|}^{(0)} = \mathbf{0}$, so the condition $\|\boldsymbol{\xi}_{\|}^{(t)}\| \geq \tau\|\boldsymbol{\xi}^{(t)}\|^2$ is violated at $t^* = 0$ for any positive $\tau$. By Lemma G.1,

$$\ell_i(\boldsymbol{\theta}^* + \boldsymbol{\xi}^{(0)}) - \ell_i(\boldsymbol{\theta}^*) \leq C_1 C_3^2 \|\boldsymbol{\xi}^{(0)}\|^4,$$

as desired; for this case, Theorem 5.1 is proved with $\rho := \rho_s$, $C := C_1 C_3^2$.

For the remaining case where $\text{span}(\{\nu_i\}_{i=1}^N) \neq \{\mathbf{0}\}$, let $H := \sum_{i=1}^N \ell_i''(\boldsymbol{\theta}^*)\nu_i\nu_i^T$, and define $\lambda_{\min}$ and $\lambda_{\max}$ to be the smallest and largest strictly positive eigenvalues of $H$, respectively. We will show that Theorem 5.1 holds with the following constant values:

$$\tau := \frac{16C_2C_4N}{\lambda_{\min}},$$

$$\rho := \frac{1}{2}\min\left\{\rho_s, \frac{\lambda_{\min}C_2}{16C_2C_5C_6N + \lambda_{\min}C_5}\right\}.$$

$$\gamma := \min\left\{\frac{8B\log 2}{\lambda_{\min}}, \frac{\lambda_{\min}B}{2\lambda_{\max}^2 E^2}\right\},$$

$$\lambda := \frac{\lambda_{\min}}{4B},$$

$$C := 16C_1(C_2\tau + C_3)^2.$$

Firstly, as we saw in the previous case, if $\|\boldsymbol{\xi}_{\|}^{(t)}\| \geq \tau\|\boldsymbol{\xi}^{(t)}\|^2$ is violated at $t^* = 0$, we immediately have

$$\ell_i(\boldsymbol{\theta}^* + \boldsymbol{\xi}^{(0)}) - \ell_i(\boldsymbol{\theta}^*) \leq C_1(C_2\tau + C_3)^2\|\boldsymbol{\xi}^{(0)}\|^4 \leq C\|\boldsymbol{\xi}^{(0)}\|^4.$$

Now suppose $\|\boldsymbol{\xi}_{\|}^{(t)}\| \geq \tau\|\boldsymbol{\xi}\|^2$ is satisfied up to some iterations, so $t^* > 0$. We will first prove that as long as $(k+1)E \leq t^*$, we have

$$\|\boldsymbol{\xi}_{\|}^{(kE+E)}\| \leq (1 - \eta\lambda)\|\boldsymbol{\xi}_{\|}^{(kE)}\|.$$

To simplify the notation, we will prove this for $k = 0$; as long as $(k+1)E \leq t^*$, the proof extends to other values of $k$.

Using Lemma G.1, we can write the gradient estimate $g^{(t)}$ at $\boldsymbol{\theta}^{(t)} = \boldsymbol{\theta}^* + \boldsymbol{\xi}^{(t)}$ as:

$$g^{(t)} = \frac{1}{B}\sum_{i \in B^{(t)}} \ell_i'(\boldsymbol{\theta}^* + \boldsymbol{\xi}^{(t)})\nabla_{\boldsymbol{\theta}} f_{\boldsymbol{\theta}^* + \boldsymbol{\xi}^{(t)}}(x_i)$$

$$= \frac{1}{B}\sum_{i \in B^{(t)}} \left(\ell_i''(\boldsymbol{\theta}^*)\nu_i^T\boldsymbol{\xi}_{\|}^{(t)} + R_i(\boldsymbol{\xi}^{(t)})\right)\left(\nu_i + \mu_i(\boldsymbol{\xi}^{(t)})\right)$$

$$= \left(\frac{1}{B}\sum_{i \in B^{(t)}} \ell_i''(\boldsymbol{\theta}^*)\nu_i\nu_i^T\right)\boldsymbol{\xi}_{\|}^{(t)} + \underbrace{\frac{1}{B}\sum_{i \in B^{(t)}} \left(\ell_i''(\boldsymbol{\theta}^*)\nu_i^T\boldsymbol{\xi}_{\|}^{(t)}\mu_i(\boldsymbol{\xi}^{(t)}) + R_i(\boldsymbol{\xi}^{(t)})(\nu_i + \mu_i(\boldsymbol{\xi}^{(t)}))\right)}_{=:\zeta^{(t)}}.$$

After the SGD update $\boldsymbol{\theta}^{(t+1)} \leftarrow \boldsymbol{\theta}^{(t)} - \eta g^{(t)}$,

$$\boldsymbol{\theta}^* + \boldsymbol{\xi}_\parallel^{(t+1)} + \boldsymbol{\xi}_\perp^{(t+1)} = \boldsymbol{\theta}^* + \boldsymbol{\xi}_\parallel^{(t)} + \boldsymbol{\xi}_\perp^{(t)} - \eta g^{(t)}$$

$$= \boldsymbol{\theta}^* + \left( I - \frac{\eta}{B} \sum_{i \in B^{(t)}} \ell_i''(\boldsymbol{\theta}^*) \nu_i \nu_i^T \right) \boldsymbol{\xi}_\parallel^{(t)} + \boldsymbol{\xi}_\perp^{(t)} - \eta \boldsymbol{\zeta}^{(t)}.$$

Since $\eta < \gamma \leq \frac{B}{\lambda_{\max}}$, $I - \frac{\eta}{B} \sum_{i \in B^{(t)}} \ell_i''(\boldsymbol{\theta}^*) \nu_i \nu_i^T$ is a positive semi-definite matrix with spectral norm at most 1. Using the projection matrix $P_\nu$, we can write

$$\boldsymbol{\xi}_\parallel^{(t+1)} = \left( I - \frac{\eta}{B} \sum_{i \in B^{(t)}} \ell_i''(\boldsymbol{\theta}^*) \nu_i \nu_i^T \right) \boldsymbol{\xi}_\parallel^{(t)} - \eta P_\nu \boldsymbol{\zeta}^{(t)}, \tag{12}$$

$$\boldsymbol{\xi}_\perp^{(t+1)} = \boldsymbol{\xi}_\perp^{(t)} - \eta (I - P_\nu) \boldsymbol{\zeta}^{(t)}. \tag{13}$$

Now, by Lemma G.1,

$$\|\boldsymbol{\zeta}^{(t)}\| \leq \frac{1}{B} \sum_{i \in B^{(t)}} \left( \|\ell_i''(\boldsymbol{\theta}^*) \nu_i^T \boldsymbol{\xi}_\parallel^{(t)} \mu_i(\boldsymbol{\xi}^{(t)})\| + \|R_i(\boldsymbol{\xi}^{(t)}) \nu_i\| + \|R_i(\boldsymbol{\xi}^{(t)}) \mu_i(\boldsymbol{\xi}^{(t)})\| \right)$$

$$\leq C_5 C_6 \|\boldsymbol{\xi}^{(t)}\| \|\boldsymbol{\xi}_\parallel^{(t)}\| + C_2 C_4 \|\boldsymbol{\xi}^{(t)}\|^2 + C_4 C_5 \|\boldsymbol{\xi}^{(t)}\|^3.$$

Under the condition that $\|\boldsymbol{\xi}_\parallel^{(t)}\| \geq \tau \|\boldsymbol{\xi}^{(t)}\|^2$, where $\tau := \frac{16 C_2 C_4 N}{\lambda_{\min}}$, and also that $\|\boldsymbol{\xi}^{(t)}\| \leq \rho \leq \frac{\lambda_{\min} C_2}{16 C_2 C_5 C_6 N + \lambda_{\min} C_5}$,

$$\|\boldsymbol{\zeta}^{(t)}\| \leq \frac{C_2 C_4}{\tau} \|\boldsymbol{\xi}_\parallel^{(t)}\| + \left( C_5 C_6 + \frac{C_4 C_5}{\tau} \right) \|\boldsymbol{\xi}^{(t)}\| \|\boldsymbol{\xi}_\parallel^{(t)}\|$$

$$\leq \left( \frac{\lambda_{\min}}{16 N} + \left( C_5 C_6 + \frac{\lambda_{\min} C_5}{16 C_2 N} \right) \|\boldsymbol{\xi}^{(t)}\| \right) \|\boldsymbol{\xi}_\parallel^{(t)}\| \leq \frac{\lambda_{\min}}{8 N} \|\boldsymbol{\xi}_\parallel^{(t)}\|.$$

From this, we can see that

$$\|\boldsymbol{\xi}_\parallel^{(t+1)}\| \leq \|\boldsymbol{\xi}_\parallel^{(t)}\| + \eta \|\boldsymbol{\zeta}^{(t)}\| \leq \left( 1 + \frac{\eta \lambda_{\min}}{8 N} \right) \|\boldsymbol{\xi}_\parallel^{(t)}\|.$$

Noting that $\eta < \gamma \leq \frac{8 B \log 2}{\lambda_{\min}}$,

$$\left( 1 + \frac{\eta \lambda_{\min}}{8 N} \right)^E \leq \left( 1 + \frac{\log 2}{E} \right)^E \leq 2,$$

so for $1 \leq t \leq E$,

$$\|\boldsymbol{\zeta}^{(t)}\| \leq \frac{\lambda_{\min}}{8 N} \|\boldsymbol{\xi}_\parallel^{(t)}\| \leq \frac{\lambda_{\min}}{8 N} \left( 1 + \frac{\log 2}{E} \right)^t \|\boldsymbol{\xi}_\parallel^{(0)}\| \leq \frac{\lambda_{\min}}{4 N} \|\boldsymbol{\xi}_\parallel^{(0)}\|.$$

Now, repeating the update rule (12) from $t = 0$ to $E - 1$, we get

$$\boldsymbol{\xi}_\parallel^{(E)} = \prod_{k=E-1}^{0} \left( I - \frac{\eta}{B} H_k \right) \boldsymbol{\xi}_\parallel^{(0)} - \eta \sum_{t=0}^{E-1} \prod_{k=E-1}^{t+1} \left( I - \frac{\eta}{B} H_k \right) P_\nu \boldsymbol{\zeta}^{(t)}, \tag{14}$$

where $H_k := \sum_{i \in B^{(k)}} \ell_i''(\boldsymbol{\theta}^*) \nu_i \nu_i^T$. We are going to bound the norm of each term. For the second term, we have

$$\left\| \sum_{t=0}^{E-1} \prod_{k=E-1}^{t+1} \left( I - \frac{\eta}{B} H_k \right) P_\nu \boldsymbol{\zeta}^{(t)} \right\| \leq \sum_{t=0}^{E-1} \|\boldsymbol{\zeta}^{(t)}\| \leq \frac{\lambda_{\min} E}{4 N} \|\boldsymbol{\xi}_\parallel^{(0)}\| = \frac{\lambda_{\min}}{4 B} \|\boldsymbol{\xi}_\parallel^{(0)}\|. \tag{15}$$

The first term is a bit tricker. Note first that

$$\prod_{k=E-1}^{0} \left( I - \frac{\eta}{B} H_k \right) = I - \frac{\eta}{B} \sum_{k=0}^{E-1} H_k + \frac{\eta^2}{B^2} \sum_{\substack{j,k \in [0,E-1] \\ j < k}} H_k H_j - \frac{\eta^3}{B^3} \sum_{\substack{i,j,k \in [0,E-1] \\ i < j < k}} H_k H_j H_i + \cdots.$$

Recall the definition $H = \sum_{i=1}^{N} \ell_i''(\boldsymbol{\theta}^*)\nu_i\nu_i^T = \sum_{k=0}^{E-1} H_k$, and that $\lambda_{\min}$ and $\lambda_{\max}$ are the minimum and maximum eigenvalues of $H$. Since $H_k$'s are positive semi-definite and $H$ is the sum of $H_k$'s, the maximum eigenvalue of $H_k$ is at most $\lambda_{\max}$. Using this,

$$\left\|\prod_{k=E-1}^{0}\left(I - \frac{\eta}{B}H_k\right)\boldsymbol{\xi}_{\|}^{(0)}\right\| \le \left(1 - \frac{\eta\lambda_{\min}}{B} + \sum_{k=2}^{E}\binom{E}{k}\left(\frac{\eta\lambda_{\max}}{B}\right)^k\right)\|\boldsymbol{\xi}_{\|}^{(0)}\|.$$

First note that for $k \in [2, E-1]$, $\binom{E}{k+1}\frac{2}{E} \le \binom{E}{k}$, because

$$\frac{2}{E} \le \frac{k+1}{E-k} = \frac{(k+1)!(E-k-1)!}{k!(E-k)!} = \frac{\binom{E}{k}}{\binom{E}{k+1}}.$$

Since $\eta < \gamma \le \frac{\lambda_{\min}B}{2\lambda_{\max}^2 E^2} \le \frac{B}{\lambda_{\max}E}$, for $k \in [2, E-1]$ we have

$$\binom{E}{k+1}\left(\frac{\eta\lambda_{\max}}{B}\right)^{k+1} \le \binom{E}{k+1}\frac{1}{E}\left(\frac{\eta\lambda_{\max}}{B}\right)^k \le \frac{1}{2}\binom{E}{k}\left(\frac{\eta\lambda_{\max}}{B}\right)^k,$$

which implies that

$$\sum_{k=2}^{E}\binom{E}{k}\left(\frac{\eta\lambda_{\max}}{B}\right)^k \le 2\binom{E}{2}\left(\frac{\eta\lambda_{\max}}{B}\right)^2 \le \frac{\eta^2 E^2 \lambda_{\max}^2}{B^2} \le \frac{\eta\lambda_{\min}}{2B}.$$

Therefore, we have

$$\left\|\prod_{k=E-1}^{0}\left(I - \frac{\eta}{B}H_k\right)\boldsymbol{\xi}_{\|}^{(0)}\right\| \le \left(1 - \frac{\eta\lambda_{\min}}{2B}\right)\|\boldsymbol{\xi}_{\|}^{(0)}\|.$$

Together with the bound on the second term (15), this shows that

$$\|\boldsymbol{\xi}_{\|}^{(E)}\| \le \left(1 - \frac{\eta\lambda_{\min}}{4B}\right)\|\boldsymbol{\xi}_{\|}^{(0)}\| = (1 - \eta\lambda)\|\boldsymbol{\xi}_{\|}^{(0)}\|,$$

which we wanted to prove.

We now have to prove that

$$\|\boldsymbol{\xi}^{(E)}\| \le \|\boldsymbol{\xi}^{(0)}\| + \eta\lambda\|\boldsymbol{\xi}_{\|}^{(0)}\|.$$

Now, repeating the update rule (13) from $t = 0$ to $E - 1$, we get

$$\boldsymbol{\xi}_{\perp}^{(E)} = \boldsymbol{\xi}_{\perp}^{(0)} - \eta\sum_{t=0}^{E-1}(I - P_\nu)\boldsymbol{\zeta}^{(t)}. \tag{16}$$

Thus, by combining equations (14) and (16),

$$\begin{aligned}
\|\boldsymbol{\xi}^{(E)}\| &= \|\boldsymbol{\xi}_{\|}^{(E)} + \boldsymbol{\xi}_{\perp}^{(E)}\| \\
&\le \left\|\prod_{k=E-1}^{0}\left(I - \frac{\eta}{B}H_k\right)\boldsymbol{\xi}_{\|}^{(0)} + \boldsymbol{\xi}_{\perp}^{(0)}\right\| + \eta\sum_{t=0}^{E-1}\left\|\prod_{k=E-1}^{t+1}\left(I - \frac{\eta}{B}H_k\right)P_\nu\boldsymbol{\zeta}^{(t)} + (I - P_\nu)\boldsymbol{\zeta}^{(t)}\right\| \\
&\le \|\boldsymbol{\xi}^{(0)}\| + \eta\sum_{t=0}^{E-1}\|\boldsymbol{\zeta}^{(t)}\| \le \|\boldsymbol{\xi}^{(0)}\| + \eta\frac{\lambda_{\min}}{4B}\|\boldsymbol{\xi}_{\|}^{(0)}\| = \|\boldsymbol{\xi}^{(0)}\| + \eta\lambda\|\boldsymbol{\xi}_{\|}^{(0)}\|.
\end{aligned}$$

It now remains to prove that $\|\boldsymbol{\xi}^{(t^*)}\| \le 2\|\boldsymbol{\xi}^{(0)}\| \le 2\rho$ at the first iteration $t^*$ that $\|\boldsymbol{\xi}_{\|}^{(t)}\| \ge \tau\|\boldsymbol{\xi}^{(t)}\|^2$ is violated. Let $k^*$ be the maximum $k$ such that $kE \le t^*$.

From what we have shown so far,

$$\|\boldsymbol{\xi}^{(k^*E)}\| \le \|\boldsymbol{\xi}^{(0)}\| + \eta\lambda\sum_{k=0}^{k^*-1}\|\boldsymbol{\xi}_{\|}^{(kE)}\|.$$

Also, for $t$ in $k^*E \le t < t^*$ the condition $\|\boldsymbol{\xi}_\parallel^{(t)}\| \ge \tau\|\boldsymbol{\xi}^{(t)}\|^2$ is satisfied, so by the same argument we have $\|\boldsymbol{\zeta}^{(t)}\| \le \frac{\lambda_{\min}}{4N}\|\boldsymbol{\xi}_\parallel^{(k^*E)}\|$ for $t \in [k^*E, t^* - 1]$. Finally, by modifying equations (14) and (16) a bit, we get

$$\|\boldsymbol{\xi}^{(t^*)}\| = \|\boldsymbol{\xi}_\parallel^{(t^*)} + \boldsymbol{\xi}_\perp^{(t^*)}\|$$

$$\le \left\| \prod_{k=t^*-1}^{k^*E} \left(I - \frac{\eta}{B}H_k\right)\boldsymbol{\xi}_\parallel^{(k^*E)} + \boldsymbol{\xi}_\perp^{(k^*E)} \right\| + \eta \sum_{t=k^*E}^{t^*-1} \left\| \prod_{k=t^*-1}^{t+1} \left(I - \frac{\eta}{B}H_k\right)P_\nu\boldsymbol{\zeta}^{(t)} + (I - P_\nu)\boldsymbol{\zeta}^{(t)} \right\|$$

$$\le \|\boldsymbol{\xi}^{(k^*E)}\| + \eta \sum_{t=k^*E}^{t^*-1} \|\boldsymbol{\zeta}^{(t)}\| \le \|\boldsymbol{\xi}^{(k^*E)}\| + \eta\frac{\lambda_{\min}}{4B}\|\boldsymbol{\xi}_\parallel^{(k^*E)}\| \le \|\boldsymbol{\xi}^{(0)}\| + \eta\lambda \sum_{k=0}^{k^*} \|\boldsymbol{\xi}_\parallel^{(kE)}\|.$$

Finally, from $\|\boldsymbol{\xi}_\parallel^{(kE+E)}\| \le (1 - \eta\lambda)\|\boldsymbol{\xi}_\parallel^{(kE)}\|$,

$$\|\boldsymbol{\xi}^{(t^*)}\| \le \|\boldsymbol{\xi}^{(0)}\| + \eta\lambda \sum_{k=0}^{k^*} (1 - \eta\lambda)^k\|\boldsymbol{\xi}_\parallel^{(0)}\| \le \|\boldsymbol{\xi}^{(0)}\| + \|\boldsymbol{\xi}_\parallel^{(0)}\| \le 2\|\boldsymbol{\xi}^{(0)}\|.$$

## H  Proof of Lemma B.1

Recall that $i_{k,j} \in [jq - q + 1, jq]$. Consider any $l < j$. Then, $i_{k,j} > lq$, so by (3) and (6), we have $a_l^1(x_{i_{k,j}}) = (-1)^{l-1}$. Similarly, if we consider $l > j$, then $i_{k,j} \le lq - q$, so it follows from (2) and (5) that $a_l^1(x_{i_{k,j}}) = (-1)^l$. This means that the entries (indexed by $(j, l)$) of $M_k$ below the diagonal are filled with $(-1)^{l-1}$, and entries above the diagonal are filled with $(-1)^l$. Thus, the matrix $M_k$ has the form

$$M_k = \begin{bmatrix} a_1^1(x_{i_{k,1}}) & 1 & -1 & \cdots & -1 & 1 & 1 \\ 1 & a_2^1(x_{i_{k,2}}) & -1 & \cdots & -1 & 1 & 1 \\ 1 & -1 & a_3^1(x_{i_{k,3}}) & \cdots & -1 & 1 & 1 \\ \vdots & \vdots & \vdots & \ddots & \vdots & \vdots & \vdots \\ 1 & -1 & 1 & \cdots & a_{p-1}^1(x_{i_{k,p-1}}) & 1 & 1 \\ 1 & -1 & 1 & \cdots & 1 & a_p^1(x_{i_{k,p}}) & 1 \end{bmatrix}.$$

To prove the first statement of Lemma B.1, consider adding the last column to every even $l$-th column and subtracting it from every odd $l$-th column. Then, this results in a matrix

$$\begin{bmatrix} a_1^1(x_{i_{k,1}}) - 1 & 2 & -2 & \cdots & -2 & 2 & 1 \\ 0 & a_2^1(x_{i_{k,2}}) + 1 & -2 & \cdots & -2 & 2 & 1 \\ 0 & 0 & a_3^1(x_{i_{k,3}}) - 1 & \cdots & -2 & 2 & 1 \\ \vdots & \vdots & \vdots & \ddots & \vdots & \vdots & \vdots \\ 0 & 0 & 0 & \cdots & a_{p-1}^1(x_{i_{k,p-1}}) - 1 & 2 & 1 \\ 0 & 0 & 0 & \cdots & 0 & a_p^1(x_{i_{k,p}}) + 1 & 1 \end{bmatrix},$$

whose columns space is the same as $M_k$. It follows from $a_j^1(x_{i_{k,j}}) \in (-1, +1)$ that $M_k$ has full column rank. This also implies that $\dim(\text{null}(M_k)) = 1$.

For the second statement, consider subtracting $(j + 1)$-th row from $j$-th row, for $j \in [p - 1]$. This results in

$$\tilde{M}_k := \begin{bmatrix} a_1^1(x_{i_{k,1}}) - 1 & 1 - a_2^1(x_{i_{k,2}}) & 0 & \cdots & 0 & 0 & 0 \\ 0 & a_2^1(x_{i_{k,2}}) + 1 & -a_3^1(x_{i_{k,3}}) - 1 & \cdots & 0 & 0 & 0 \\ 0 & 0 & a_3^1(x_{i_{k,3}}) - 1 & \cdots & 0 & 0 & 0 \\ \vdots & \vdots & \vdots & \ddots & \vdots & \vdots & \vdots \\ 0 & 0 & 0 & \cdots & a_{p-1}^1(x_{i_{k,p-1}}) - 1 & 1 - a_p^1(x_{i_{k,p}}) & 0 \\ 1 & -1 & 1 & \cdots & 1 & a_p^1(x_{i_{k,p}}) & 1 \end{bmatrix},$$

which has the same null space as $M_k$. Consider a nonzero vector $\nu \in \text{null}(\tilde{M}_k)$, i.e., $\tilde{M}_k \nu = \mathbf{0}$. Let $\nu_l$ denote the $l$-th component of $\nu$. One can see that $\nu_1, \ldots, \nu_p$ are not all zero, because if $\nu_{p+1}$ is the only nonzero component, $\tilde{M}_k \nu = (0, 0, \ldots, 0, \nu_{p+1})^T \neq \mathbf{0}$.

Assume without loss of generality that $\nu_1$ is strictly positive. Note that $a_1^1(x_{i_{k,1}}) - 1$ and $1 - a_2^1(x_{i_{k,2}})$ are both nonzero and the signs of $a_1^1(x_{i_{k,1}}) - 1$ and $1 - a_2^1(x_{i_{k,2}})$ are opposite. Then it follows from $(a_1^1(x_{i_{k,1}}) - 1)\nu_1 + (1 - a_2^1(x_{i_{k,2}}))\nu_2 = 0$ that $\nu_2$ is also strictly positive. Similarly, $a_2^1(x_{i_{k,2}}) + 1$ and $-a_3^1(x_{i_{k,3}}) - 1$ are both nonzero and have opposite signs, so $\nu_3 > 0$. Proceeding this way up to $\nu_p$, we can see that all $\nu_l$, $l \in [p]$, are strictly positive.

# I    Proof of Lemma G.1

We begin by introducing more definitions. For a matrix $A \in \mathbb{R}^{m \times n}$, let $\text{vec}(A) \in \mathbb{R}^{mn}$ be its vectorization, i.e., columns of $A$ concatenated as a long vector. Given matrices $A$ and $B$, let $A \otimes B$ denote their Kronecker product. Throughout the proof, we use $\boldsymbol{\theta}$ and $\boldsymbol{\xi}$ to denote the concatenation of vectorizations of all the parameters $(\boldsymbol{W}^l, \boldsymbol{b}^l)_{l=1}^L$ and perturbations $(\boldsymbol{\Delta}^l, \boldsymbol{\delta}^l)_{l=1}^L$:

$$
\boldsymbol{\theta} := \begin{bmatrix} \text{vec}(\boldsymbol{W}^L) \\ \boldsymbol{b}^L \\ \text{vec}(\boldsymbol{W}^{L-1}) \\ \boldsymbol{b}^{L-1} \\ \vdots \\ \text{vec}(\boldsymbol{W}^1) \\ \boldsymbol{b}^1 \end{bmatrix}, \quad \boldsymbol{\xi} := \begin{bmatrix} \text{vec}(\boldsymbol{\Delta}^L) \\ \boldsymbol{\delta}^L \\ \text{vec}(\boldsymbol{\Delta}^{L-1}) \\ \boldsymbol{\delta}^{L-1} \\ \vdots \\ \text{vec}(\boldsymbol{\Delta}^1) \\ \boldsymbol{\delta}^1 \end{bmatrix}. \tag{17}
$$

In Section 2, we defined $a^l(x_i)$ to denote output of the $l$-th hidden layer when the network input is $x_i$. In order to make the dependence of parameters more explicit, we will instead write $a_{\boldsymbol{\theta}}^l(x_i)$ in this section. Also, for $l \in [L-1]$, define

$$
D_{\boldsymbol{\theta}}^l(x_i) := W^L J_{\boldsymbol{\theta}}^{L-1}(x_i) W^{L-1} \cdots W^{l+1} J_{\boldsymbol{\theta}}^l(x_i) \in \mathbb{R}^{1 \times d_l}, \tag{18}
$$

and for convenience in notation, let $D_{\boldsymbol{\theta}}^L(x_i) := 1$. It can be seen from standard matrix calculus that

$$
\begin{bmatrix} \nabla_{\boldsymbol{W}^l} f_{\boldsymbol{\theta}}(x_i) & \nabla_{\boldsymbol{b}^l} f_{\boldsymbol{\theta}}(x_i) \end{bmatrix} = D_{\boldsymbol{\theta}}^l(x_i)^T \begin{bmatrix} a_{\boldsymbol{\theta}}^{l-1}(x_i)^T & 1 \end{bmatrix}, \tag{19}
$$

for all $l \in [L]$. Vectorizing and concatenating these partial derivatives results in

$$
\nabla_{\boldsymbol{\theta}} f_{\boldsymbol{\theta}}(x_i) = \begin{bmatrix} \begin{bmatrix} a_{\boldsymbol{\theta}}^{L-1}(x_i) \\ 1 \end{bmatrix} \\ \begin{bmatrix} a_{\boldsymbol{\theta}}^{L-2}(x_i) \\ 1 \end{bmatrix} \otimes D_{\boldsymbol{\theta}}^{L-1}(x_i)^T \\ \vdots \\ \begin{bmatrix} x_i \\ 1 \end{bmatrix} \otimes D_{\boldsymbol{\theta}}^1(x_i)^T \end{bmatrix}. \tag{20}
$$

In order to prove the lemma, we first have to quantify how perturbations on the global minimum affect outputs of the hidden layers and the network. Let $\boldsymbol{\theta}^* := (\boldsymbol{W}_*^l, \boldsymbol{b}_*^l)_{l=1}^L$ be the memorizing global minimum, and let $(\boldsymbol{\Delta}^l, \boldsymbol{\delta}^l)_{l=1}^L$ be perturbations on parameters, whose vectorization $\boldsymbol{\xi}$ satisfies $\|\boldsymbol{\xi}\| \leq \rho_c$. Then, for all $l \in [L-1]$, define $\tilde{a}_{\boldsymbol{\theta}^* + \boldsymbol{\xi}}^l(\cdot)$ to denote the amount of perturbation in the $l$-th hidden layer output:

$$
\tilde{a}_{\boldsymbol{\theta}^* + \boldsymbol{\xi}}^l(x_i) := a_{\boldsymbol{\theta}^* + \boldsymbol{\xi}}^l(x_i) - a_{\boldsymbol{\theta}^*}^l(x_i).
$$

It is easy to check that

$$
\tilde{a}_{\boldsymbol{\theta}^* + \boldsymbol{\xi}}^1(x_i) = J_{\boldsymbol{\theta}^*}^1(x_i)(\boldsymbol{\Delta}^1 x_i + \boldsymbol{\delta}^1),
$$
$$
\tilde{a}_{\boldsymbol{\theta}^* + \boldsymbol{\xi}}^l(x_i) = J_{\boldsymbol{\theta}^*}^l(x_i) \left( \boldsymbol{\Delta}^l a_{\boldsymbol{\theta}^*}^{l-1}(x_i) + \boldsymbol{\delta}^l + (\boldsymbol{W}_*^l + \boldsymbol{\Delta}^l) \tilde{a}_{\boldsymbol{\theta}^* + \boldsymbol{\xi}}^{l-1}(x_i) \right).
$$

Similarly, let $\tilde{f}_{\boldsymbol{\theta}^*+\boldsymbol{\xi}}(\cdot)$ denote the amount of perturbation in the network output. It can be checked that

$$\tilde{f}_{\boldsymbol{\theta}^*+\boldsymbol{\xi}}(x_i) := f_{\boldsymbol{\theta}^*+\boldsymbol{\xi}}(x_i) - f_{\boldsymbol{\theta}^*}(x_i) = \boldsymbol{\Delta}^L a_{\boldsymbol{\theta}^*}^{L-1}(x_i) + \boldsymbol{\delta}^L + (\boldsymbol{W}_*^L + \boldsymbol{\Delta}^L)\tilde{a}_{\boldsymbol{\theta}^*+\boldsymbol{\xi}}^{L-1}(x_i).$$

One can see that $\tilde{a}_{\boldsymbol{\theta}^*+\boldsymbol{\xi}}^1(x_i)$ only contains perturbation terms that are first-order in $\boldsymbol{\xi}$: $J_{\boldsymbol{\theta}^*}^1(x_i)(\boldsymbol{\Delta}^1 x_i + \boldsymbol{\delta}^1)$. However, the order of perturbation accumulates over layers. For example,

$$\tilde{a}_{\boldsymbol{\theta}^*+\boldsymbol{\xi}}^2(x_i) = J_{\boldsymbol{\theta}^*}^2(x_i)\left(\boldsymbol{\Delta}^2 a_{\boldsymbol{\theta}^*}^1(x_i) + \boldsymbol{\delta}^2 + (\boldsymbol{W}_*^2 + \boldsymbol{\Delta}^2)J_{\boldsymbol{\theta}^*}^1(x_i)(\boldsymbol{\Delta}^1 x_i + \boldsymbol{\delta}^1)\right)$$
$$= \underbrace{J_{\boldsymbol{\theta}^*}^2(x_i)\left(\boldsymbol{\Delta}^2 a_{\boldsymbol{\theta}^*}^1(x_i) + \boldsymbol{\delta}^2 + \boldsymbol{W}_*^2 J_{\boldsymbol{\theta}^*}^1(x_i)(\boldsymbol{\Delta}^1 x_i + \boldsymbol{\delta}^1)\right)}_{\text{first-order perturbation}} + \underbrace{J_{\boldsymbol{\theta}^*}^2(x_i)\boldsymbol{\Delta}^2 J_{\boldsymbol{\theta}^*}^1(x_i)(\boldsymbol{\Delta}^1 x_i + \boldsymbol{\delta}^1)}_{\text{second-order perturbation}},$$

so $\tilde{a}_{\boldsymbol{\theta}^*+\boldsymbol{\xi}}^2(x_i)$ contains 1st–2nd order perturbations. Similarly, $\tilde{a}_{\boldsymbol{\theta}^*+\boldsymbol{\xi}}^l(x_i)$ has terms that are 1st–$l$-th order in $\boldsymbol{\xi}$, and $\tilde{f}_{\boldsymbol{\theta}^*+\boldsymbol{\xi}}(\cdot)$ perturbation terms from 1st order to $L$-th order.

Using the definition of $D_{\boldsymbol{\theta}}^l(x_i)$ from Eq (18), the collection of first order perturbation terms in $\tilde{f}_{\boldsymbol{\theta}^*+\boldsymbol{\xi}}(\cdot)$ can be written as

$$\tilde{f}_{\boldsymbol{\theta}^*+\boldsymbol{\xi}}^1(x_i) := \boldsymbol{\Delta}^L a_{\boldsymbol{\theta}^*}^{L-1}(x_i) + \boldsymbol{\delta}^L + \boldsymbol{W}_*^L J_{\boldsymbol{\theta}^*}^{L-1}(x_i)(\boldsymbol{\Delta}^{L-1} a_{\boldsymbol{\theta}^*}^{L-2}(x_i) + \boldsymbol{\delta}^{L-1}) + \cdots$$
$$= \sum_{l=1}^L D_{\boldsymbol{\theta}^*}^l(x_i)(\boldsymbol{\Delta}^l a_{\boldsymbol{\theta}^*}^{l-1} + \boldsymbol{\delta}^l) \overset{(a)}{=} \nabla_{\boldsymbol{\theta}} f_{\boldsymbol{\theta}^*}(x_i)^T \boldsymbol{\xi} \overset{(b)}{=} \nu_i^T \boldsymbol{\xi}_{\|}$$

where (a) is an application of Taylor expansion of $f_{\boldsymbol{\theta}}(x_i)$ at $\boldsymbol{\theta}^*$, which can also be directly checked from explicit forms of $\boldsymbol{\xi}$ (17) and $\nabla_{\boldsymbol{\theta}} f_{\boldsymbol{\theta}^*}(x_i)$ (20). Equality (b) comes from the definition of $\boldsymbol{\xi}_{\perp}$ that $\boldsymbol{\xi}_{\perp} \perp \nu_i$. We also define the collection of higher order perturbation terms:

$$\tilde{f}_{\boldsymbol{\theta}^*+\boldsymbol{\xi}}^{2+}(x_i) := \tilde{f}_{\boldsymbol{\theta}^*+\boldsymbol{\xi}}(x_i) - \tilde{f}_{\boldsymbol{\theta}^*+\boldsymbol{\xi}}^1(x_i).$$

Now, from the definition of memorizing global minima, $\ell_i'(\boldsymbol{\theta}^*) = 0$ for all $i \in [N]$. Since $\ell_i$ is three times differentiable, Taylor expansion of $\ell_i(\cdot)$ at $\boldsymbol{\theta}^*$ gives

$$\ell_i(\boldsymbol{\theta}^* + \boldsymbol{\xi}) - \ell_i(\boldsymbol{\theta}^*) = \frac{1}{2}\ell_i''(\boldsymbol{\theta}^*)(\tilde{f}_{\boldsymbol{\theta}^*+\boldsymbol{\xi}}(x_i))^2 + \frac{1}{6}\alpha_i(\tilde{f}_{\boldsymbol{\theta}^*+\boldsymbol{\xi}}(x_i))^3,$$

where $\alpha_i = \ell'''(f_{\boldsymbol{\theta}^*}(x_i) + \beta_i \tilde{f}_{\boldsymbol{\theta}^*+\boldsymbol{\xi}}(x_i); y_i)$ for some $\beta_i \in [0,1]$. For small enough $\rho_s$, $\tilde{f}_{\boldsymbol{\theta}^*+\boldsymbol{\xi}}(x_i)$ is small enough and bounded, so there exists a constant $C_1$ such that

$$\ell_i(\boldsymbol{\theta}^* + \boldsymbol{\xi}) - \ell_i(\boldsymbol{\theta}^*) \leq C_1(\tilde{f}_{\boldsymbol{\theta}^*+\boldsymbol{\xi}}(x_i))^2$$

for all $i \in [N]$. There also are constants $C_2 := \max_{i \in [N]} \|\nu_i\|$ and $C_3$ such that

$$|\tilde{f}_{\boldsymbol{\theta}^*+\boldsymbol{\xi}}^1(x_i)| \leq C_2\|\boldsymbol{\xi}_{\|}\|, \quad \text{and} \quad |\tilde{f}_{\boldsymbol{\theta}^*+\boldsymbol{\xi}}^{2+}(x_i)| \leq C_3\|\boldsymbol{\xi}\|^2$$

for all $i \in [N]$, therefore

$$\ell(f_{\boldsymbol{\theta}^*+\boldsymbol{\xi}}(x_i); y_i) - \ell(f_{\boldsymbol{\theta}^*}(x_i); y_i) \leq C_1(C_2\|\boldsymbol{\xi}_{\|}\| + C_3\|\boldsymbol{\xi}\|^2)^2$$

holds for all $i \in [N]$, as desired.

Now, consider the Taylor expansion of $\ell_i'$ at $f_{\boldsymbol{\theta}^*}(x_i)$. Because $\ell_i'$ is twice differentiable and $\ell_i'(\boldsymbol{\theta}^*) = 0$,

$$\ell_i'(\boldsymbol{\theta}^* + \boldsymbol{\xi}) = \ell''(\boldsymbol{\theta}^*)\tilde{f}_{\boldsymbol{\theta}^*+\boldsymbol{\xi}}(x_i) + \frac{1}{2}\hat{\alpha}_i(\tilde{f}_{\boldsymbol{\theta}^*+\boldsymbol{\xi}}(x_i))^2$$

$$= \ell''(f_{\boldsymbol{\theta}^*}(x_i); y_i)\tilde{f}_{\boldsymbol{\theta}^*+\boldsymbol{\xi}}^1(x_i) + \underbrace{\ell''(f_{\boldsymbol{\theta}^*}(x_i); y_i)\tilde{f}_{\boldsymbol{\theta}^*+\boldsymbol{\xi}}^{2+}(x_i) + \frac{1}{2}\hat{\alpha}_i(\tilde{f}_{\boldsymbol{\theta}^*+\boldsymbol{\xi}}(x_i))^2}_{=:R_i(\boldsymbol{\xi})}$$

$$= \ell''(f_{\boldsymbol{\theta}^*}(x_i); y_i)\nu_i^T \boldsymbol{\xi}_{\|} + R_i(\boldsymbol{\xi}), \tag{21}$$

where $\hat{\alpha}_i = \frac{1}{2}\ell'''(f_{\boldsymbol{\theta}^*}(x_i) + \hat{\beta}_i \tilde{f}_{\boldsymbol{\theta}^*+\boldsymbol{\xi}}(x_i); y_i)$ for some $\hat{\beta}_i \in [0,1]$. The remainder term $R_i(\boldsymbol{\xi})$ contains all the perturbation terms that are 2nd-order or higher, so there is a constant $C_4$ such that

$$|R_i(\boldsymbol{\xi})| \leq C_4\|\boldsymbol{\xi}\|^2$$

holds for all $i \in [N]$.

In a similar way, we can see from Eq (20) that we can express $\nabla_{\boldsymbol{\theta}} f_{\boldsymbol{\theta}^* + \boldsymbol{\xi}}(x_i)$ as the sum of $\nu_i :=$ $\nabla_{\boldsymbol{\theta}} f_{\boldsymbol{\theta}^*}(x_i)$ plus the perturbation $\mu_i(\boldsymbol{\xi})$:

$$\nabla_{\boldsymbol{\theta}} f_{\boldsymbol{\theta}^* + \boldsymbol{\xi}}(x_i) = \nu_i + \mu_i(\boldsymbol{\xi}),$$

where $\mu_i(\boldsymbol{\xi})$ contains all the perturbation terms that are 1st-order or higher. So, there exists a constant $C_5$ such that

$$\|\mu_i(\boldsymbol{\xi})\| \leq C_5 \|\boldsymbol{\xi}\|$$

holds for all $i \in [N]$.