[Reviews · NeurIPS 2019]

Reviewer 1



The paper focuses on the memorization capacity of ReLU networks. For 3-layer FNNs the authors tighten the bounds by some carefully constructed parameters to fit the dataset perfectly. As they point out, some related works focus on shallow networks and it could be hard to apply the analysis to more practical networks. The authors provide a lower bound for memorization capacity of deeper networks, by using a set of 3-layer FNNs as blocks in the proofs. To the best of my knowledge, the result is novel. The paper is structured clearly and easy to follow. I enjoyed reading the paper.

Reviewer 2



Clear presentation. Results are original and contribute to our understanding of ReLu networks. They make a significant advance to existing literature.

Reviewer 3



The paper investigates the problem of expressiveness in neural networks w.r.t. finite samples, and improves current bound of O(N) to \Theta(\sqrt N) w.r.t. a two hidden layer network, where N is the sample size (showing both a lower and an upper bound). The authors also show an upper bound for classification, a corollary of which is that a three hidden layer network with hidden layers of sized 2k-2k-4k can perfectly classify ImageNet. Moreover, they show that if the overall sum of hidden nodes in a ResNet is of order N/d_x, where d_x is the input dimension then again the network can perfectly realize the data. Lastly, an analysis is given showing batch SGD that is initialized close to a global minimum will come close to a point with value significantly smaller than the loss in the initialization (though a convergence guarantee could not be given). The paper is clear and easy to follow for the most part, and conveys a feeling that the authors did their best to make the analysis as thorough and exhausting as possible, providing results for various settings.

[Author Response · NeurIPS 2019]

We would like to thank the reviewers for their valuable comments and encouraging feedback. Below, we address the
concerns raised in the order they appeared.

**Reviewer 1.**
**Q. Could the authors elaborate a little on $r_j$ (line 245)? [...] specific because of the construction in appendix E?**
As seen in Figure 2, the construction in Appendix E partitions the dataset into several groups of size $N_1, N_2, \ldots, N_m$
and uses the construction in Theorem 3.1 as building blocks to fit each group of data. The additional requirement
of $r_j$ nodes corresponds to circle and diamond nodes in Figure 2, because we need to propagate input information
using a hidden node up to layer $l_m$ (the last "building block" that fits $N_m$ points), and propagate output information
using $d_y$ hidden nodes starting from layer $l_1 + 2$ (after the first group of $N_1$ points are fitted). This is why we need
$r_j = d_y \mathbf{1}\{j > 1\} + \mathbf{1}\{j < m\}$ extra nodes in Proposition 3.4. The reason why we need only one hidden node for input
information is because we down-project $x_i$'s onto a line. This requirement of $r_j$ is indeed specific to the construction.

**Reviewer 2.**
We appreciate the reviewer's positive comments about our submission.

**Reviewer 3.**
**Q. Omega notation throughout the paper is erroneous?**
We agree that our choice of this notation might cause confusion. The reason why we used $\Omega(\sqrt{N})$ for sufficiency
statements, e.g., in line 2, was that anything of greater order than $\sqrt{N}$ (e.g., $N$) could also memorize $N$ data points.
However, we agree that it is erroneous to say that $\Omega(\sqrt{N})$ is **necessary**, because this may sound as if $N$ nodes are also
necessary for memorization. We will correct the $\Omega$ notation to $\Theta$ throughout the paper. Thank you for the correction.

**Q. Comparison to [Soudry and Carmon, 2016]?**
Thank you for pointing out a relevant paper. We believe that their results are not directly comparable to ours because
there are a few important differences in the problem settings. The biggest difference is that [SC16] consider a setting
where there is a multiplicative "dropout noise" at each hidden node and each data point. At $i$-th node of $l$-th layer, the
slope of the activation function for the $n$-th data point is either $\epsilon_{i,l}^{(n)} \cdot 1$ (if input is positive) or $\epsilon_{i,l}^{(n)} \cdot s$ (if negative, $s \neq 0$),
where $\epsilon_{i,l}^{(n)}$ is the multiplicative random (e.g., Gaussian) dropout noise. Their theorem statements hold for "almost every
realization" of these dropout noise factors, so heavily depend on their particular model. In contrast, our setting is free
of these noise terms, and hence corresponds to a **specific** realization of such $\epsilon_{i,l}^{(n)}$'s. The discussion after Theorem 5
(the multiple hidden layer result) of [SC16] suggests that their proof crucially depends on the "except measure zero"
argument on some of the noise terms $\epsilon_{\cdot,L-1}^{(\cdot)}$, hence making our results not directly comparable to theirs.

**Q. Theorem 5.1: The statement was a bit difficult to parse. [...] in the theorem and their magnitude.**
The exact values of positive constants can be found in Appendix G, and they are dependent on a number of terms, such
as the number of data points $N$, batch size $B$, the radius $\rho_s$ of a ball in which the slopes of activation don't change,
the Taylor expansions of loss $\ell(f_{\boldsymbol{\theta}^*}(x_i); y_i)$ and network output $f_{\boldsymbol{\theta}^*}(x_i)$ around the memorizing global minimum $\boldsymbol{\theta}^*$,
maximum and minimum strictly positive eigenvalues of $H = \sum_{i=1}^N \ell''(f_{\boldsymbol{\theta}^*}(x_i); y_i)\nu_i \nu_i^T$, where $\nu_i = \nabla_{\boldsymbol{\theta}} f_{\boldsymbol{\theta}^*}(x_i)$. We
will make sure to add detailed explanation and an improved theorem statement in the next revision.

**Q. If I understand correctly, [...] the initialization must be significantly close in the first place?**
It is indeed true that in the worst case the point found at $t^*$ can be farther away from $\boldsymbol{\theta}^*$, and also that our theorem
requires the initialization to be close to the global minimum. However, an initialization that is $\epsilon$-close (in Euclidean
distance) to the global minimum has empirical risk $O(\epsilon^2)$ (shown by Lemma G.1). Thus, in terms of risk value, the
initialization is not necessarily as close to the global minimum compared to the point found at $t^*$, which achieves $O(\epsilon^4)$
risk. We'd like to highlight that one can start off at a $O(\epsilon^2)$-risk point and quickly find a $O(\epsilon^4)$-risk point.

**Q. Comparison to [Zhong et al., 2017]?**
Thanks for bringing up this point. We would like to emphasize that the settings are quite different, so one cannot
make direct comparisons. [ZSJ+17] consider 1-hidden-layer networks with $\pm 1$-valued weights at the output layer,
with an implicit assumption that network width is smaller than input dimension. Input $x_i$ is Gaussian and output $y_i$ is
generated by a "teacher" network. In comparison, we consider arbitrary datasets and networks, under a mild assumption
(especially for overparametrized networks) that the network can memorize the data. We are happy to add more detailed
comparisons to our next revision.

**Q. Note that in their analysis, Zhong et al. show that the strong convexity [...] this technique?**
It seems difficult to show strong convexity in our case. For example, for ReLU networks, if we scale one layer by $\alpha$ and
the next layer by $\alpha^{-1}$, we get exactly the same network. This means that for ReLU, the global minimum always has a
direction in which the risk value does not change, hence strong convexity cannot hold at global minima. The key to
[ZSJ+17]'s result is that they fix the output layer parameters to $\pm 1$, and only consider hidden layer parameters.

[Meta-Review · NeurIPS 2019]

The paper analyses the memorization capacity of small RELU networks. The topic is timely, and the results would be of interest to a wide audience. The reviewers found the paper well written and were also satisfied with the authors response. However, please do take the time to address their comments and revise what is necessary in the final version.